# Sufficiency analysis of estrogen responsive enhancers using synthetic activators

Matthew Ginley-Hidinger[1,2] ⓘ, Julia B Carleton[1,3], Adriana C Rodriguez[1,3], Kristofer C Berrett[1,3], Jason Gertz[1,3] ⓘ

**Multiple regulatory regions bound by the same transcription factor have been shown to simultaneously control a single gene's expression. However, it remains unclear how these regulatory regions combine to regulate transcription. Here, we test the sufficiency of promoter-distal estrogen receptor α-binding sites (ERBSs) for activating gene expression by recruiting synthetic activators in the absence of estrogens. Targeting either dCas9-VP16(10x) or dCas9-p300(core) to ERBS induces H3K27ac and activates nearby expression in a manner similar to an estrogen induction, with dCas9-VP16(10x) acting as a stronger activator. The sufficiency of individual ERBSs is highly correlated with their necessity, indicating an inherent activation potential that is associated with the binding of RNA polymerase II and several transcription factors. By targeting ERBS combinations, we found that ERBSs work independently to control gene expression when bound by synthetic activators. The sufficiency results contrast necessity assays that show synergy between these ERBSs, suggesting that synergy occurs between ERBSs in terms of activator recruitment, whereas directly recruiting activators leads to independent effects on gene expression.**

## Introduction

Gene expression enhancers are genomic loci that act in *cis* to control a distal gene's expression level. There are two orders of magnitude more predicted enhancers in the human genome than gene promoters (Roadmap Epigenomics Consortium et al, 2015), indicating that many mammalian genes are regulated by multiple enhancers. Analysis of 3D genome architecture (ENCODE Project Consortium, 2012) and the expression of enhancer RNAs (Andersson et al, 2014) corroborate the idea that the average human gene is regulated by the combined action of many enhancers. Functional studies into enhancer combinations have found that enhancers can work together in an additive/independent (Fujioka et al, 1999; Bender et al, 2012), synergistic (Lam et al, 2015; Torbey et al, 2018), or

redundant (Hong et al, 2008; Osterwalder et al, 2018) manner, indicating that enhancers can combine to regulate gene expression in complex and diverse ways.

There are two different approaches to functionally perturb enhancers to study enhancer function: tests of necessity and tests of sufficiency. Deleting or inhibiting enhancer function tests the necessity of an enhancer for endogenous gene expression. The sufficiency of enhancer sequences can be studied by ectopic reporter assays (Catarino & Stark, 2018). However, testing only enhancer sequences does not uncover how enhancers act in their endogenous environment. To determine whether an enhancer region is sufficient within the genomic context, enhancers must be directly activated in an unbiased way. Most studies of enhancer function and combinatorics involve genetic deletion of the region(s) of interest. Genetic deletion, along with CRISPR interference-based approaches (Gilbert et al, 2013; Thakore et al, 2015; Fulco et al, 2016; Korkmaz et al, 2016; Carleton et al, 2017), test the necessity of a genomic region or combination of genomic regions for regulatory activity. Regulatory element screens that test sufficiency have identified shared and unique regulatory elements in comparison with screens for necessity, indicating that necessity and sufficiency are complementary assays (Klann et al, 2017). Testing the sufficiency of an enhancer by direct activation provides additional insight into the innate capabilities of an enhancer, the requirements of enhancer function, and whether genomic properties at enhancer regions associate with an enhancer's potential for modulating transcription. However, testing the sufficiency of genomic regulatory regions in their native context is less commonly undertaken than necessity.

Reporter assays, in which a plasmid containing a potential enhancer sequence controls the expression of a reporter gene, are one way of testing enhancer sufficiency (Catarino & Stark, 2018). However, this technique does not analyze an enhancer within its native chromatin context and is likely missing information that might be critical for the ability of an enhancer to regulate gene expression, such as epigenetic marks, interaction of enhancers with specific promoters, and properties of adjacent genomic regions (Cunningham et al, 2018). CRISPR activation (CRISPRa) is a recently developed tool for activating gene expression from specific regions of the genome. CRISPRa involves the fusion of a catalytically dead

[1]Huntsman Cancer Institute, University of Utah, Salt Lake City, UT, USA  [2]Department of Bioengineering, University of Utah, Salt Lake City, UT, USA  [3]Department of Oncological Sciences, University of Utah, Salt Lake City, UT, USA

Correspondence: jay.gertz@hci.utah.edu

Cas9 protein (dCas9) to an activation domain. Different activation domains can be fused to dCas9, including the transcriptional activation domain of VP16 (Cheng et al, 2013) and the enzymatic core domain of p300 (Hilton et al, 2015). VP16 is a herpes simplex virus transcription factor (TF) which recruits a variety of host factors, including general TFs, mediator, and histone acetyltransferases (Hirai et al, 2010). p300 is a histone acetyltransferase linked with activation of many genomic regions (Delvecchio et al, 2013). CRISPRa provides a strategy for turning on specific genes when targeted to promoter regions with gRNAs (Cheng et al, 2013; Perez-Pinera et al, 2013; Chavez et al, 2015). CRISPRa can also be targeted to distal regulatory regions to test their sufficiency in promoting gene expression (Hilton et al, 2015; Klann et al, 2017; Thormann et al, 2018); however, very few distal regulatory regions have been interrogated in this manner, and it is unclear how targeting CRISPRa to combinations of enhancers will impact gene expression.

Many enhancers are controlled by the activity of inducible TFs. Estrogen receptor α (ER) is a steroid hormone receptor which only has gene regulatory activity when it is bound by estrogens. ER acts mostly as an activating TF, binding to thousands of genomic loci and regulating hundreds of genes (Gertz et al, 2012, 2013). Most genes that are up-regulated by estrogen have multiple ER-bound sites nearby and we have previously found evidence of collaboration between ER-bound sites in regulating the gene expression response to estrogens (Carleton et al, 2017). Using a CRISPRi-based approach, enhancer interference (Enhancer-i), we found synergistic and hierarchical relationships involving ER-bound sites. These relationships were discovered by measuring the necessity of each ER-bound site individually and in combination. It is unclear whether estrogen receptor–binding site (ERBS) collaboration is a property of ER or dependent on the genomic properties of ERBSs, necessitating an investigation of ERBS sufficiency in the genomic context.

Here, we use CRISPRa to target combinations of regulatory regions that are normally bound by ER (Fig 1A). By targeting these regions in the absence of estrogens, we sought to determine if CRISPRa synthetic activators could recreate the transcriptional response to estrogens. We find that dCas9-VP16(10x) fusion can recreate most of the estrogen response at the four genes tested, although dCas9-p300(core) was not as effective. Targeting CRISPRa to individual regulatory regions and combinations of loci uncovered an additive/independent relationship between sites, in contrast to our previous necessity findings. Our results indicate that ER binding to neighboring enhancers works in a synergistic fashion, but synthetic activators directly recruited to loci normally bound by ER work independently to regulate gene expression.

## Results

### Targeting CRISPRa to loci normally bound by ER can mimic transcriptional responses to estrogens

To determine if synthetic activators are sufficient to drive an estrogen-like transcriptional response, we evaluated two CRISPRa fusion proteins: dCas9-VP16(10x) (Cheng et al, 2013), which is commonly used at promoters, and dCas9-p300(core) (Hilton et al, 2015),

which can activate gene expression from enhancers. Each fusion was expressed from identical expression vectors (see the Materials and Methods section, Fig S1A) to directly compare their ability to activate gene expression from distal regulatory regions normally bound by ER. We first targeted dCas9-VP16(10x) and dCas9-p300(core) to the *IL1RN* promoter, a gene which can be highly activated by CRISPRa (Cheng et al, 2013; Perez-Pinera et al, 2013; Hilton et al, 2015), and observed a similar level of activation for *IL1RN* with both dCas9 fusion constructs in Ishikawa cells, an endometrial cancer cell line (Fig S1B).

The dCas9 fusions were targeted to a pool of ERBSs to determine if the estrogen response could be recapitulated with CRISPRa. We chose a set of 12 ERBSs that were within 125 kb of four genes that are normally responsive to estrogen. *MMP17* has two upstream and one downstream ERBSs, *CISH* has three downstream ERBSs, and *FHL2* and *HES2* both have three upstream ERBSs (Fig 1B). We targeted the CRISPRa fusions to the 12 ERBSs simultaneously and measured gene expression of the target genes in the absence of estrogens and, therefore, the absence of ER binding. We observed significant gene expression activation by dCas9-VP16(10x) at all four genes tested, whereas activation by dCas9-p300(core) was significant at three genes (*MMP17*, *CISH*, and *HES2*) (Fig 1C–F). For comparison, we also targeted our constructs to the promoter regions of these four genes and found that targeting ERBSs often results in equal or greater activation than targeting the promoter (Fig 1C–F). dCas9-VP16(10x) was consistently a stronger activator than dCas9-p300(core). The level of gene expression driven by dCas9-VP16(10x) was somewhat correlated with the fold change in gene expression seen with a 17β-estradiol (E2) induction at this set of four genes (r = 0.8) (Fig S1C). These results demonstrate that CRISPRa targeted to ERBSs can mimic the activation seen in an E2 transcriptional response.

To determine if the activation potential is specific to ERBSs, we targeted dCas9-VP16(10x) and dCas9-p300(core) to a total of six regions surrounding *MMP17*, *CISH*, and *FHL2* that are at most 8 kb away from ERBSs discussed above (or the transcriptional start site [TSS] in the case of FHL2-B) (Fig 1B). Regions with low DNase I hypersensitivity signal (Gertz et al, 2013) were chosen, to limit the probability that the locus was an active regulatory region controlled by other TFs. As *HES2* is in a highly active region with several DNase I hypersensitive sites and histone H3 lysine 27 acetylation positive loci, multiple nearby genes, and many TF-binding events, we were unable to choose sites that were not potential regulatory regions at this locus. In choosing adjacent regions, we aimed to keep the distance to the TSS similar without being too close to the ER-bound site. We observed some TFs binding to the chosen adjacent regions, notably at CISH-A (Table S3). When targeting the ERBS-adjacent sites, we did not observe significant activation over the control of targeting the *IL1RN* promoter (Fig 1C–E). The inability of ERBS-adjacent regions to regulate gene expression when bound by synthetic activators indicates specificity when testing sufficiency of regulatory regions with CRISPRa and differences in activation potential between ERBSs and nearby non-ERBSs.

### dCas9 activator fusions can target precise genomic loci and induce histone acetylation

To test whether CRISPRa is successfully targeted to the intended genomic regions surrounding our genes of interest, we conducted a chromatin immunoprecipitation followed by sequencing (ChIP-seq)

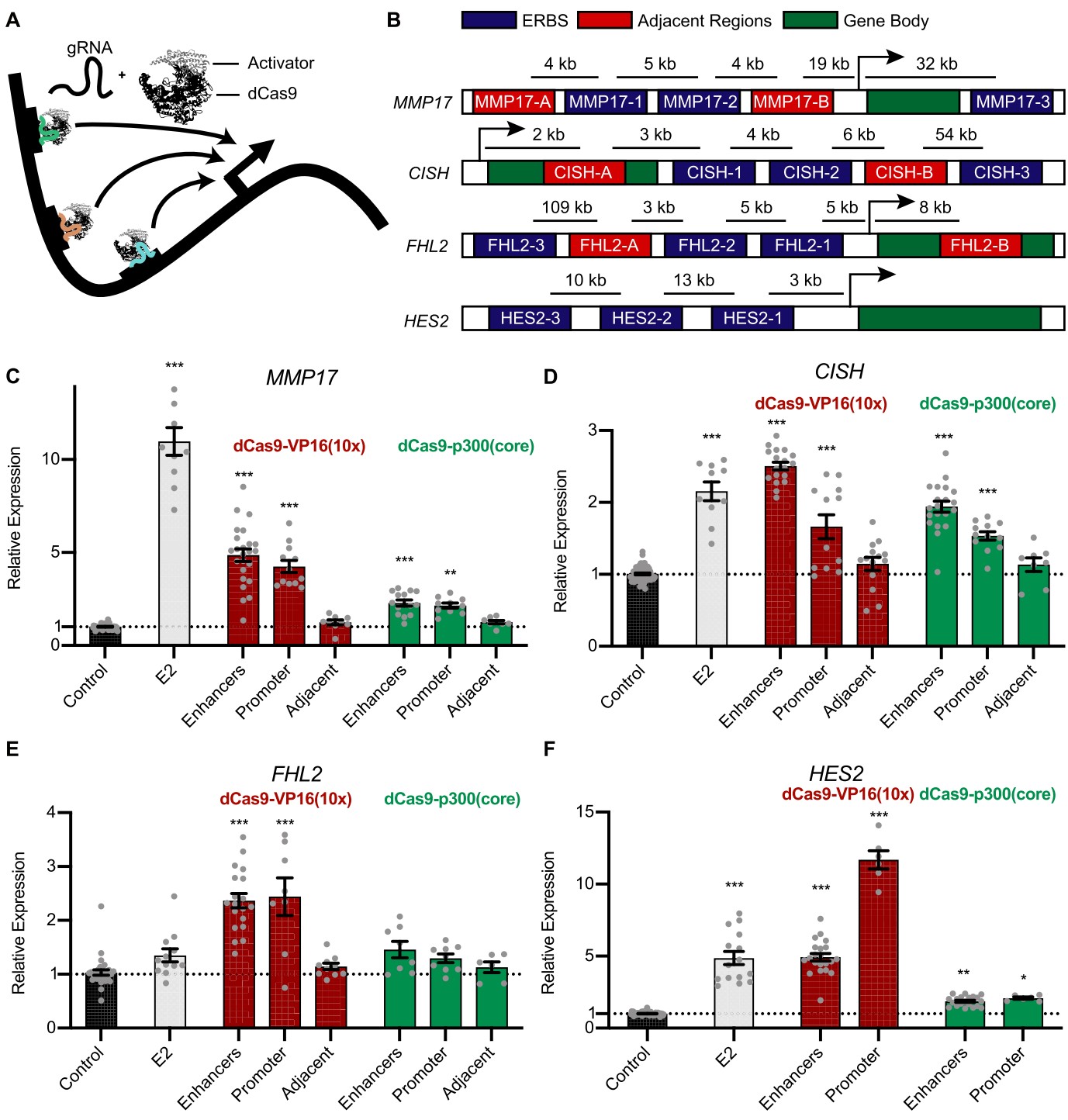

**Figure 1. Targeting multiple ERBSs with synthetic activators can activate gene expression.**
**(A)** Cartoon depicting the targeting of multiple ERBSs in combination to study combinatorial effects on gene expression. **(B)** Relative locations of ERBSs (blue), ERBS-adjacent regions (red), and genes (green) tested in this study. **(C–F)** Targeting all 12 ERBSs in combination with dCas9-vp16(10x) (red) or dCas9-p300(core) (green) activated gene expression at *MMP17* (C), *CISH* (D), *FHL2* (E), and *HES2* (F) to levels that are comparable with an 8-h E2 treatment (light gray). Targeting all ERBSs had significantly higher activation than targeting ERBS-adjacent regions, which is not significantly different than controls that target the *IL1RN* promoter. Error bars represent SEM. *P*-values are calculated with respect to control using a one-way ANOVA with Dunnett's multiple comparisons (\**P*-value < 0.05, \*\**P*-value < 0.01, \*\*\**P*-value < 0.001).

experiment using an antibody that recognizes an HA epitope tag on dCas9. dCas9-VP16(10x) was targeted to 19 loci, consisting of 12 ERBSs, 6 ERBS-adjacent regions, and the *IL1RN* promoter (Fig 1B). At

all of these loci, we observed a distinct HA (dCas9) signal at the targeted site when compared with nontargeted controls (Figs 2A and B, and S2A–C). In addition, dCas9-p300(core) was successfully

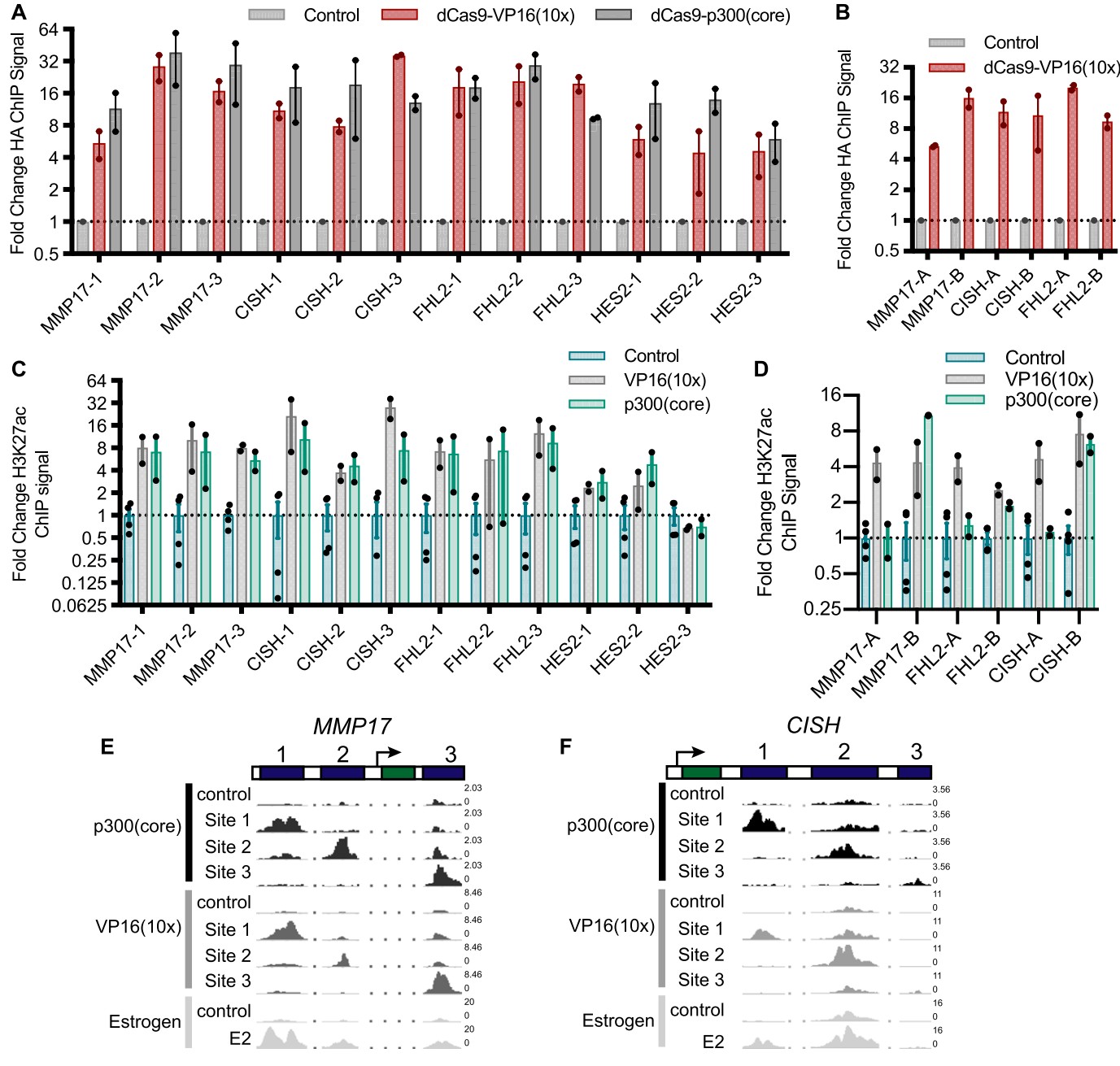

**Figure 2. Targeting dCas9-activator constructs is specific and induces H3K27ac.**
**(A, B)** The relative HA ChIP-seq signal, an epitope tag on dCas9, is shown for all targeted ERBSs (A) and adjacent regions (B) and compared with non-targeted controls. Points indicate individual replicates and error bars represent SEM. **(C)** The fold change induction of H3K27ac ChIP-seq signal across all targeted loci shows no significant difference between dCas9-p300(core) and VP16(10x). **(D)** H3K27ac was induced at all adjacent regions by dCas9-VP16(10x) and at three of six adjacent regions by dCas9-p300(core). Points indicate individual replicates and error bars represent SEM. **(E, F)** Browser tracks of H3K27ac induced by targeting ERBSs with dCas9-p300(core), dCas9-VP16(10x), or an 8-h E2 treatment are shown at *MMP17* (E) and *CISH* (F). Numbers on the right of the tracks indicate the track height in non-normalized reads per million.

targeted to all 12 ERBSs (Fig 2A). Successful targeting of dCas9-VP16(10x) to ERBS-adjacent regions indicates the lack of activation from these regions is based on genomic properties of the adjacent regions and not a result of the targeting efficiency.

Histone 3 lysine 27 acetylation (H3K27ac) is a histone modification found at active regulatory regions and is directly deposited by p300 (Raisner et al, 2018). We, therefore, performed ChIP-seq with an antibody that recognizes H3K27ac to determine if the CRISPRa

fusions were able to cause H3K27ac at targeted sites. For 18 of 19 targeted loci, we observed increased H3K27ac (Figs 2C and D, and S2C–G). Notably, at HES2-1 and HES2-3, there is significant baseline H3K27ac present, possibly because of the binding of other TFs to these sites (Table S2). For ERBSs, the patterns of acetylation are similar to E2-induced H3K27ac (Figs 2E and F, and S2D and E). We observed similar fold changes in H3K27ac when using dCas9-VP16(10x) and when using dCas9-p300(core) targeted to ERBSs

(Figs 2C, E, and F, and S2D and E; *P*-value = 0.108, paired *t* test). This result is in contrast to the greater gene activation induced by dCas9-VP16(10x), indicating that histone acetylation of a distal regulatory element is not fully predictive of target gene activation. Consistent with the idea that H3K27ac by itself is not sufficient to drive maximal expression, we observed H3K27ac at the adjacent regions even though they were unable to induce gene expression (Fig S2F). In addition, we conducted ChIP-seq with an antibody for RNA Polymerase II (RNAPII) and found limited RNAPII recruitment to ERBSs by dCas9-p300(core) or dCas9-VP16(10x) (Fig S2H), although we saw RNAPII recruitment by dCas9-VP16(10x) at the *IL1RN* promoter (Fig S2C).

### dCas9-VP16(10x) activates gene expression from individual ERBSs

Because targeting dCas9-VP16(10x) to all ERBSs simultaneously resulted in gene activation of all genes, we next sought to determine if targeting individual ERBSs with dCas9-VP16(10x) is sufficient to increase gene expression. At the *MMP17* locus, all three ERBSs activated gene expression above the control level when targeted individually (Fig 3A, pairwise *P*-values in Table S1). Targeting MMP17-1 resulted in the highest induced expression, MMP17-2 exhibited the weakest activation and MMP17-3 led to an intermediate change in *MMP17* expression. When targeting the ERBS surrounding *CISH*, CISH-1 and CISH-2 induced a similar level of activation, whereas CISH-3 did not result in activation (Fig 3B). The *FHL2* gene was induced strongest by FHL2-1. *FHL2* was also activated by FHL2-2 and FHL2-3, but to a lower level (Fig 3C). At *HES2*, we observed a high level of activation from HES2-1 and slight activation from HES2-2 and HES2-3 (Fig 3D). Targeting dCas9-p300(core) to individual ERBSs resulted in a lower, but correlated, activation in comparison to dCas9-VP16(10x) (r = 0.633, Fig S4A–D), indicating that the relative strength of enhancers may be independent of the synthetic activator used, whereas absolute strength can be controlled by the strength of the synthetic activator.

To ensure that the activation observed from individual sites is specific to the targeted site, we tried to activate expression in previously derived Ishikawa lines with homozygous deletion of the targeted ERBSs for the *MMP17* and *CISH* sites (Carleton et al, 2017). In each case, no detectable gene activation was observed in the deletion lines, indicating specificity in targeting individual ERBSs (Fig S3A and B). We theorized the observed differences in activation at individual sites may be due to biases in dCas9 targeting, preferential H3K27ac or differences in RNAPII recruitment. However, we found no significant correlation between expression activation and fold change HA ChIP-seq signal, fold change H3K27ac ChIP-seq signal or fold change RNAPII ChIP-seq signal (Fig S3C–E). The ability to activate gene expression from several individual enhancers adds to a growing body of literature showing that a gene's expression can be controlled by multiple regulatory regions. Furthermore, the unique level of activation that results from targeting individual enhancers suggests that enhancers do not contribute equally to gene activation, even when bound by a synthetic activator.

### Enhancers bound by synthetic activators work independently to regulate transcription

Based on the observation that multiple ERBSs nearby each gene were capable of activating gene expression when targeted individually

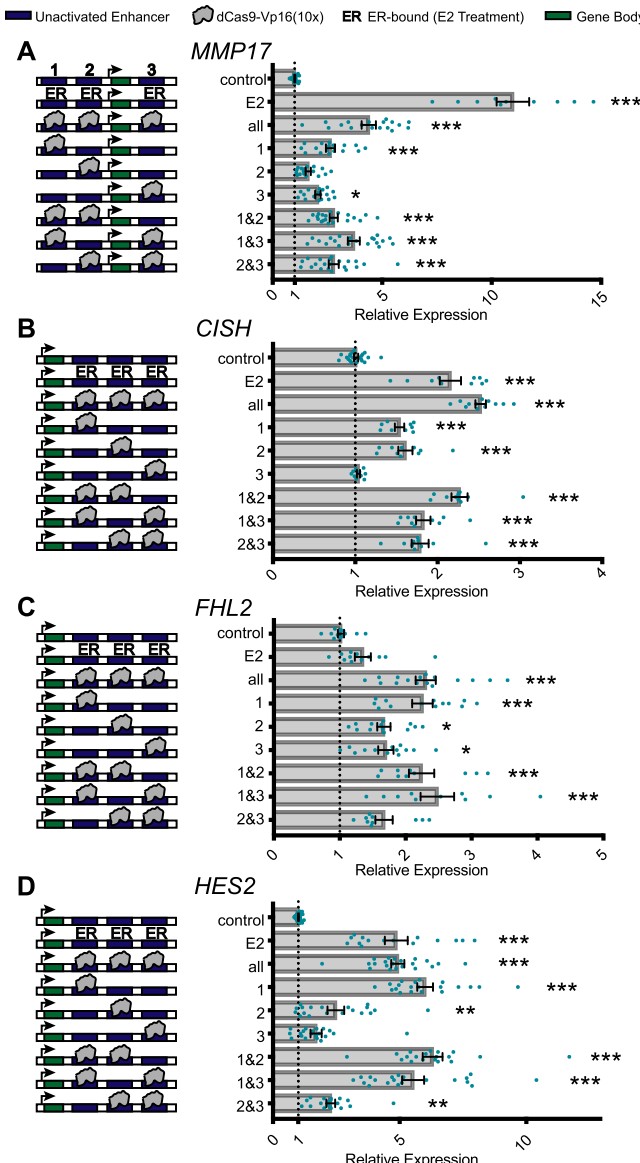

**Figure 3. Activation of gene expression by targeting CRISPRa to combinations of enhancers.**
**(A–D)** (left) The combination of ERBSs targeted by dCas9-VP16(10x) or bound by ER upon E2 treatment are shown in a schematic. **(A–D)** (right) The relative fold change in expression as measured by qRT-PCR, when compared with control cells with guides targeting the *IL1RN* promoter, was determined for each combination of ERBSs. Each data point is shown as a blue dot; error bars represent SEM. Pairwise log2 ratios and significance levels are given in Table S1 (Pairwise *t* test *P*-values comparing to control: *** < 0.001, ** < 0.01, * < 0.05).

by dCas9-VP16(10x), we investigated how synthetic activator bound enhancers collaborate to control gene expression by targeting pairs of ERBSs. In general, ERBSs appeared to combine independently when simultaneously targeted. For example, CISH-1 and CISH-2 are the two strongest individual activators of *CISH* and each increase gene expression to ~40% of maximum observed activation, whereas the combination of CISH-1 and CISH-2 increases gene expression to 80% (Fig 3B). A similar pattern was observed for *MMP17* (Fig 3A), whereas *FHL2* (Fig 3C) and *HES2* (Fig 3D) exhibit subadditive effects that may represent saturation.

To quantitatively understand the interactions between ERBSs when bound by a synthetic activator, we created a thermodynamic model of RNAPII recruitment (as a surrogate for transcription) by ERBSs (Shea & Ackers, 1985; Buchler et al, 2003). To describe the differences in gene expression seen from our combinatorial activation studies, we fit relative energy parameters to an abstracted model of combinatorial synthetic activation (Fig 4A). The model included four sets of interactions: (1) interactions between dCas9-VP16(10x) and targeted ERBSs (Fig 4, red), (2) interactions between ERBS-bound synthetic activators (Fig 4, green), (3) interactions between ERBS-bound synthetic activators and RNAPII (Fig 4, blue), and (4) interactions between RNAPII and a gene's promoter (Fig 4, purple). This model assumes that the probability of RNAPII binding is proportional to a gene's expression. We used the correlation between the probability of RNAPII being bound and gene expression data from targeting combinations of ERBSs (Fig 3) to fit model parameters. We ran the parameter optimization with many random starts, then selected parameters that fit gene expression levels reasonably well (within 0.1 of optimal) (see the Materials and Methods section). We consequently observed a range of parameters that were

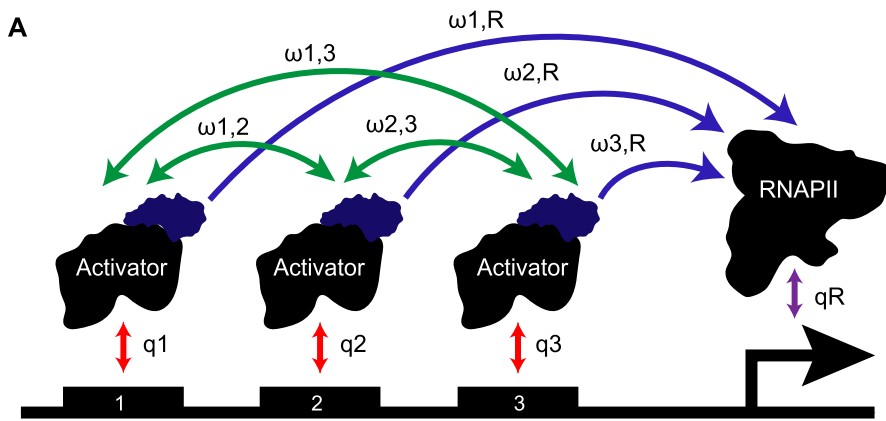

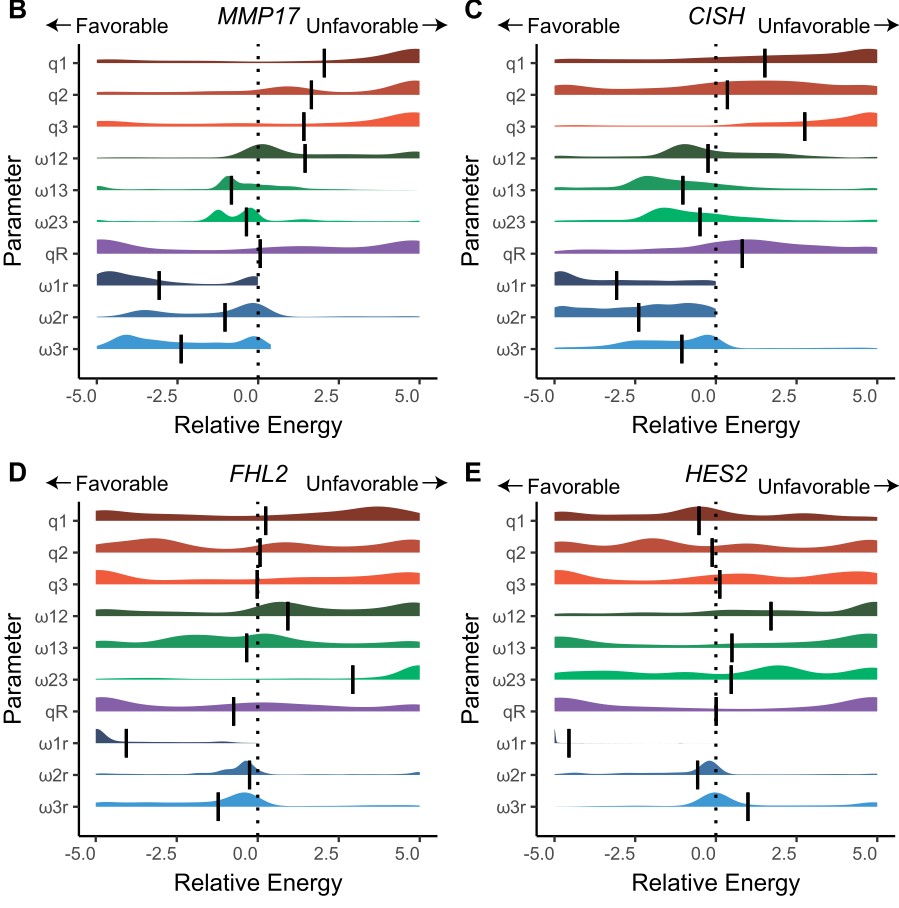

**Figure 4. Thermodynamic modeling reveals little cooperativity between synthetic activator-bound ERBSs.**
**(A)** Schematic showing the set of modeled parameters. **(B–E)** Parameters were fit to gene expression data for *MMP17*, *CISH*, *FHL2*, and *HES2* (from Fig 3). Plots show the distribution of fitted parameters. Parameter sets were selected if the modeled data correlated with gene expression data within 0.1 of an optimal correlation. Vertical bars represent the mean.

locally optimal (Fig 4B–E). In some cases, we observed multimodal parameter distributions, which is likely due to parameters balancing each other in different ways, leading to multiple local optima.

In this model, the activation observed from targeting individual loci is largely captured by the interaction terms between RNAPII and synthetic activators (parameter ranges shown in blue in Fig 4), where more favorable (more negative) interactions are indicative of more gene activation. For example, at *FHL2*, site FHL2-1 has the largest impact on expression and the most favorable interaction with RNAPII, whereas FHL2-2 and FHL2-3 have modest effects on expression and slightly favorable RNAPII-synthetic activator interactions (Fig 4D). In these models, the interaction between synthetic activators and RNAPII can be balanced by differential recruitment of the synthetic activators to the ERBSs (shown in red in Fig 4). For example, CISH-1 and CISH-2 both activate gene expression to similar levels, but CISH-1 is modeled with the synthetic activator more strongly recruiting RNAPII, whereas CISH-2 is modeled as binding the synthetic activator with more efficiency (Fig 4C). The relationship between ERBSs in the sufficiency experiments is best captured by the interaction terms between synthetic activators bound to ERBSs (parameter ranges shown in green in Fig 4). For all studied genes, we do not see strong cooperativity between ERBSs. For *MMP17* and *CISH*, we observed relatively neutral interactions between ERBSs (Fig 4B and C). The best fits for *FHL2* and *HES2* were mostly competitive models where certain ERBSs inhibit others (Fig 4D and E). This may result from a limit in how much gene activation can be driven by the synthetic activators. For example, dCas9-VP16(10x) at HES2-1 results in a similar gene expression level as targeting all ERBSs surrounding *HES2* simultaneously. Even though targeting of HES2-2 or HES2-3 has some activity in isolation, they are unable to increase expression beyond HES2-1 targeting (Fig 3D). This is captured in the model as unfavorable interactions between HES2-1 and the other ERBSs. In general, the lack of cooperativity in these models supports the conclusion that these sites work independently to activate gene expression when targeted with dCas9-VP16(10x).

It is possible that the observed independence of synthetically activated ERBSs is due to the strength of dCas9-VP16(10x), which may override the subtleties of enhancer synergy. We, therefore, used thermodynamic models to analyze activation by a weaker activator, dCas9-p300(core), at the three genes which can be activated by this construct. Again, we observed an independent relationship at *MMP17* (Fig S4E). At *CISH* and *HES2*, we observed bimodal distributions of some interaction parameters, where a subset of fitted models indicate synergy between enhancers; however, most model fits indicate independence between enhancers when bound by dCas9-p300(core) (Fig S4F and G). The presence of multiple local optima might be due to low levels of activation and lower signal-to-noise ratios. Overall, we see that sites activated by dCas9-p300(core) do not display strong patterns of cooperativity, consistent with the results from dCas9-VP16(10x) targeting.

**Comparison between necessity and sufficiency of ERBSs**

We previously assessed the necessity of the nine ERBSs nearby *CISH*, *MMP17*, and *FHL2* in producing a transcriptional response to

estrogen using Enhancer-i (Carleton et al, 2017). For Enhancer-i, ERBSs were inhibited using dCas9 with both the SIN3A interacting domain (SID) of MAD1 and the Krüppel associated box (KRAB) domain. At these three genes, we found the predominant ERBSs for activating gene expression when testing sufficiency were the same as when testing necessity (e.g., MMP17-1 and FHL2-1). To normalize the relative impact of targeting individual ERBSs for each gene, we calculated the z-score of relative expression when targeting an individual ERBSs compared with targeting the other individual ERBSs for that gene. We observed a strong correlation between the relative necessity, as measured by Enhancer-i (and validated by genetic deletion), and sufficiency, as measured by dCas9-VP16(10x) targeting (r = 0.840, Fig 5A). The consistent importance of individual ERBSs, in terms of sufficiency and necessity, suggests that each ERBS has a native activation potential that is unique to the site.

To determine what genomic traits are predictive of enhancer strength, we looked at a set of eight possible predictors of activation by both dCas9-p300(core) and dCas9-VP16(10x). The strongest correlated variables to dCas9-VP16(10x) and dCas9-p300(core) activation included the number of TFs present at each site, the base level of RNAPII ChIP-seq signal, and the base level of H3K27ac ChIP-seq signal (Figs 5D and S5E–H). Using a multiple regression-based method, the relative importance of each variable was calculated (see the Materials and Methods section) (Groemping, 2006). This revealed that the best predictor of dCas9-VP16(10x) activation was the number of TFs present at the site, whereas the best predictor for dCas9-p300(core)-mediated activation was the base amount of RNAPII present at each site (Fig 5E). The number of TFs bound to each site was determined by totaling the number of ChIP-seq peaks that overlap ERBSs and adjacent regions using publicly available ChIP-seq data (ENCODE Project Consortium, 2012) for 18 different TFs in Ishikawa cells (Tables S2 and S3). Although no TF was bound solely to strongly activated sites, we found certain TFs, such as TCF12 and ZBTB7A, were bound more often to enhancers that exhibited strong activation when targeted with synthetic activators (Table S2). Overall, these data suggest that dCas9-VP16(10x) and dCas9-p300(core) may have different requirements for target gene activation.

Although the necessity and sufficiency of individual ERBSs was consistent, combinations of ERBSs behaved differently when comparing necessity and sufficiency. We previously reported synergy between ERBSs when testing necessity. To quantitatively compare ERBS interactions between CRISPRa and Enhancer-i, we thermodynamically modeled the Enhancer-i data using the same model as activation with the difference being a site was defined as "active" if it was not blocked (i.e., untargeted by SID-dCas9-KRAB). In contrast to the models based on CRISPRa, we observed the expected cooperative interactions between pairs of ERBSs: MMP17-1 and MMP17-2, CISH-1 and CISH-2, and FHL2-1 and FHL2-2 (Fig S5A–C). When comparing parameters between the Enhancer-i and CRISPRa-derived models, it is clear that most parameter estimates are consistent, except for ERBS interaction terms between activated enhancers (Figs 5B and C, and S5D). These results are consistent with synergy in gene regulation occurring between ERBSs when ER is binding to the enhancers, whereas ERBSs that are instead bound by synthetic activators act independently on gene expression. Therefore, synergy likely occurs in the recruitment of ER and potentially cofactors to an ERBS, whereas targeting an activator directly to an ERBS does not require synergy with neighboring ERBSs.

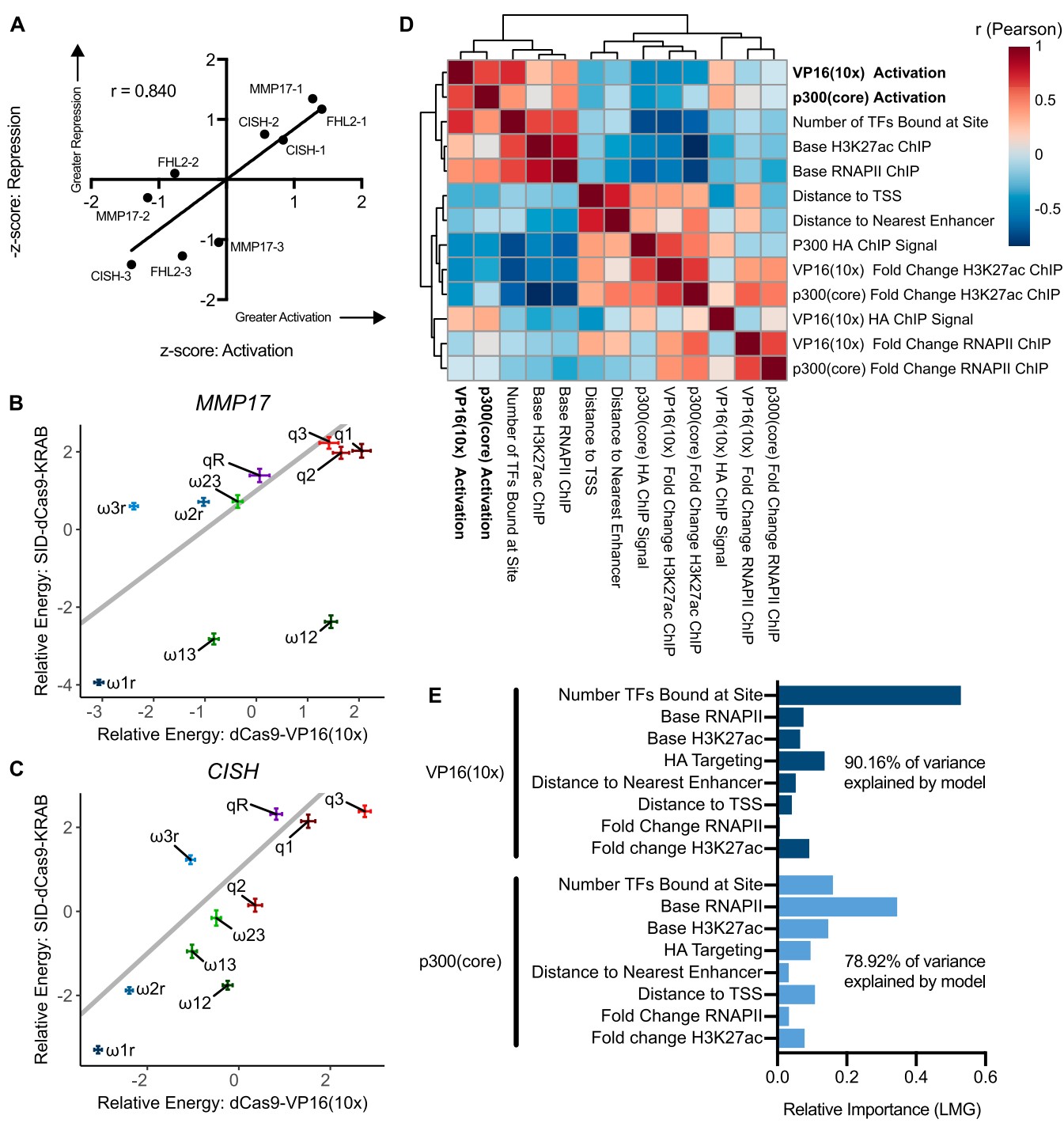

**Figure 5. Sufficiency-necessity comparison shows similar results at individual sites, but differences in cooperation between sites.**
**(A)** Scatter plot shows relative expression, as measured by z-score, for both activation and interference at individual ERBSs. Z-scores were negated for interference so that a higher score is associated with greater necessity. **(B, C)** Scatter plots show the parameters of thermodynamic models derived from CRISPRa and Enhancer-i for *MMP17* (B) and *CISH* (C). Parameters are shown as mean ± 95% confidence intervals. **(D)** Clustered correlation matrix showing Pearson correlations between potential predictors and both dCas9-VP16(10x) activation and dCas9-p300(core) activation. **(E)** Analysis of relative importance of predictors for activation using the Lindeman, Merenda, and Gold method (see the Materials and Methods section). Importance is normalized to sum to one.

## Discussion

CRISPR-based gene activation is a unique platform for interrogating the sufficiency of gene expression enhancers within the native genomic context. In this study, we applied variations of the approach to distal regulatory elements normally bound by ER, using cells that were not exposed to estrogens and therefore had negligible ER activity. Targeting dCas9, fused to either the enzymatic

core of p300 or 10 copies of VP16, to 12 ERBSs surrounding four genes simultaneously activated transcription to levels that mimicked an E2 transcriptional response. We also observed H3K27ac deposition at all sites targeted, in a pattern similar to that caused by E2 treatment. Regions adjacent to ERBSs do not activate gene expression when targeted, suggesting that certain loci have the potential to impact gene expression when locally activated and others do not.

In agreement with the idea of inherent activation potential of an enhancer, we also found that the sufficiency of an ERBS, as measured by dCas9-VP16(10x) gene activation from the site, is correlated with the necessity of the site, as measured by SID(4x)-dCas9-KRAB interference (Carleton et al, 2017). This observation supports the notion that inducible enhancers have an intrinsic ability to activate gene expression to a certain magnitude. Analysis of potential genomic factors that may determine activation potential revealed that the best predictor of dCas9-VP16(10x) activation is the number of other TFs present at the site. A battery of other TFs may help stabilize dCas9-VP16(10x) or may perform orthogonal roles in enhancer maturation. This finding indicates that the underlying DNA sequence may be important for regulation by synthetic activators as they may be working together with other TFs. For dCas9-p300(core), the most important predictor is the base amount of RNAPII already present at the site, which could point to an important difference between the synthetic activators in their ability to recruit RNAPII, as VP16 has been shown to recruit basal TFs (Hirai et al, 2010). The unique predictors from this analysis suggest that different modes of activation are used by dCas9-VP16(10x) and dCas9-p300(core), which warrants further study.

We tested two dCas9 fusions and found that dCas9-VP16(10x) activates genes to a higher level than dCas9-p300(core). The fusions caused similar levels of H3K27ac to be deposited at targeted loci, suggesting that histone acetylation is not the only event that impacts transcription and that VP16 is likely contributing to gene activation in other ways. This could be because VP16 recruits a host of cofactors, including basal TFs and mediator, in addition to histone acetyltransferases (Hirai et al, 2010). These interactions allow VP16 to more directly assemble the transcriptional machinery, whereas p300-induced acetylation may be limited by other methods of transcriptional control, such as protein recruitment. The superior performance of dCas9-VP16(10x) may also be specific to the ERBS that we targeted, as dCas9-p300(core) has been shown to be more effective than VP16(4x) at other loci (Hilton et al, 2015), or it could be explained by the extra copies of VP16 in the 10x fusion.

Although the necessity and sufficiency of individual ERBSs were well correlated, the manner in which they combine to regulate gene expression was very different when tested with Enhancer-i and CRISPRa. In necessity experiments, pairs of ERBSs showed synergistic behavior. For example, *CISH* was not estrogen responsive unless both CISH-1 and CISH-2 were active. In the sufficiency experiments, ERBSs combined in a mostly independent/additive fashion, although some combinations appear sub-additive and may approach a saturating level of activation for the gene. One explanation for the independence between enhancers when bound by synthetic activators is that the synthetic activators are so strong that they override more subtle regulatory events that require synergy. However, we do not believe this to be the case because an E2 induction leads to higher expression levels than either activator,

especially dCas9-p300(core). The contrast in how ERBSs combine to regulate transcription in the two experimental approaches suggests that the synergy between ERBSs likely occurs in *cis*, where the recruitment of ER and its cofactors, such as histone acetyltransferases (Hanstein et al, 1996; Shang et al, 2000), is synergistic between ERBSs. However, if the cofactors are directly recruited to the ERBSs, as is the case with CRISPRa, then synergy no longer occurs. In this model, synergy occurs when ERBSs influence one another before activation. Then, once ERBSs are activated through the binding of TFs and cofactors, ERBSs communicate with the target gene independently. There are important caveats to consider when using these synthetic activators, including the possible recruitment of cofactors that do not normally bind to a particular enhancer or potential interference with TF binding by dCas9. However, we believe that the comparison between enhancer activation and Enhancer-i has shed light on the consistent importance of individual sites as well as a key difference in how enhancers work together when bound by different transcriptional activators.

## Materials and Methods

### dCas9 construct generation

The Addgene 48227 plasmid (a gift from Rudolf Jaenisch) (Cheng et al, 2013) containing dCas9-VP16(10x) with a P2A linker and neomycin resistance gene was used for dCas9-VP16(10x) as well as the starting point for our dCas9-p300(core) construct. The p300(core) insert was obtained via PCR from the Addgene 61357 (gift from Charles Gersbach) (Hilton et al, 2015) plasmid using primers (Table S4), which also added AscI and ClaI restriction enzyme sites. Both the p300(core) PCR product and the dCas9-VP16(10x) plasmid were digested by AscI and ClaI, removing the C-terminal VP16(10x) from the plasmid, and subsequently ligated together. Constructs were verified via Sanger sequencing (Table S4) (Genewiz).

### gRNA design

gRNAs were designed and cloned as previously described (Carleton et al, 2017). Four gRNA oligos were designed for each target region and pooled before Gibson cloning. gRNA plasmids were then pooled equally by site into three pools, such that each pool contained gRNAs targeted to one ERBS near each of the four genes studied as listed below. An adjacent pool was also created containing all gRNAs targeted to non-ERBS regions. Pools were created using equal mixtures of gRNA plasmids by mass.

> Pool 1: MMP17-2, CISH-1, FHL2-2, HES2-3.
> Pool 2: MMP17-1, CISH-2, FHL2-3, HES2-1.
> Pool 3: MMP17-3, CISH-3, FHL2-1, HES2-2.
> Adjacent Pool: MMP17-A, MMP17-B, CISH-A, CISH-B, FHL2-A, FHL2-B.

The sequences of the gRNAs targeting the ERBSs surrounding *MMP17*, *CISH*, *FHL2*, *HES2*, adjacent regions, and promoters can be found in Table S5. Previously described gRNAs targeting the *IL1RN* promoter (Perez-Pinera et al, 2013) were used as controls.

## Cell culture and transfection

A human endometrial adenocarcinoma cell line, Ishikawa (Sigma-Aldrich), was used for ChIP-seq and gene expression experiments. Ishikawa cells were cultured in Roswell Park Memorial Institute (RPMI) medium (Gibco) supplemented with 10% fetal bovine serum (Gibco) and 1% penicillin–streptomycin (Gibco) and incubated at 37°C with 5% $CO_2$. The cells were transferred to hormone-depleted media (phenol red-free RPMI [Gibco] with 10% charcoal-dextran stripped fetal bovine serum [Sigma-Aldrich]), at least 5 d before transfection by gRNA and dCas9 fusion plasmids. Ishikawa deletion lines were previously created and verified. All deletions are homozygous deletions selected through single cell cloning. Deletions were cultured in the same conditions as parental Ishikawa cells, again being transferred to hormone-depleted media at least 5 d before transfection.

The cells were transfected using the FuGENE HD Reagent (Promega) according to the manufacturer's protocol for unlisted cells. dCas9 fusions (dCas9-VP16(10x) or dCas9-p300(core)) plasmids were transfected at a mass ratio of 3:2 to pooled gRNA plasmids. Mass ratio of dCas9 fusions to tomato reporter plasmid (Addgene 30530, gift from Gerhart Ryffel) was 6:1. Plasmid solutions were prepared in Opti-MEM.

## ChIP-seq

Cells were grown in hormone-depleted media for 5–7 d and then plated in 15-cm dishes at 8.5 million cells per dish. Cells were transfected 1 d after plating as described above with 42.75 $\mu$g total plasmid. Approximately 40 h after transfection, the media was changed to fresh hormone-depleted media containing 1 $\mu$g/ml puromycin and 300 $\mu$g/ml G418 to select for cells transfected with both plasmids. Chromatin was harvested 72 h posttransfection. ChIP was performed as previously described (Reddy et al, 2009). The antibodies used were HA (16B12; BioLegend), H3K27ac (pAb Cat. no. 39133; Active Motif), and RNAPII (ab5408 [4H8]; Abcam). Libraries were sequenced on the Illumina HiSeq 2500 as single-end 50-base pair reads and aligned to hg19 using bowtie with parameters -m 1 –t –best -q -S -l 32 -e 80 -n 2 (Langmead et al, 2009). Duplicates were removed from binary alignment map (BAM) files using samtools rmdup with modifier flag–s for single-end reads (Li et al, 2009). Counts were generated using bedtools coverage (Quinlan & Hall, 2010) and normalized for total read depth. For H3K27ac ChIP-seq and RNAPII ChIP-seq, the counts were then normalized to the average read depth within all overlapping peaks for a given antibody. H3K27ac levels before normalization show the same activation trends, with different baseline levels (Fig S2G). For HA ChIP-seq, there were not any overlapping peaks that were not targeted; therefore, counts were normalized to the mean of five control regions with background signal: *CTCF* promoter (chr16:67,594,830–67,596,830), *TBP* promoter (chr6: 170,862,978–170,864,978), *SF3B4* promoter (chr1:149,898,675–149,900,675), and *TRIM28* promoter (chr19:59,054,414–59,056,414). Fold change in ChIP-seq signal was calculated as the ratio of a targeted region's normalized counts versus the same region's normalized counts in a non-targeted control. ChIP-seq data for H3K27ac (DMSO and following an 8-h E2 induction) have been previously published and are accessible at Gene Expression Omnibus (GSE99906) (Carleton et al, 2017).

## Gene expression analysis

Before gene expression analysis using qRT-PCR, Ishikawa cell lines were grown in hormone-depleted media for 5–8 d before being plated in 24-well plates at 60,000–100,000 cells per well. The cells were transfected 1 d after plating with 550 ng total DNA per well. Approximately 40 h after transfection, the media was changed to hormone-depleted media containing 1 $\mu$g/ml puromycin and 300 $\mu$g/ml G418 to select for successfully transfected cells. E2 inductions were performed by adding 10 nM E2 to media 64 h posttransfection. 72 h posttransfection, the cells were lysed using Buffer RLT Plus (QIAGEN) with 1% $\beta$-mercaptoethanol (Sigma-Aldrich). RNA was isolated using the ZR-96-well Quick-RNA kit (Zymo Research) and quantified using either RiboGreen (Life Technologies) with an EnVision plate reader (PerkinElmer) or with a Qubit 2.0 (Life Technologies).

qRT-PCR was conducted using the Power SYBR Green RNA-to-CT 1-Step Kit (Life Technologies). 50 ng of starting material was mixed into a 20-$\mu$l reaction volume. A CFX Connect light cycler (Bio-Rad) was used to perform a 30-min cDNA synthesis at 48°C followed by a 10-min enzyme activation at 95°C and 40 cycles of 15 s at 95°C and 1 min at 60°C. qRT-PCR primers were added at a final concentration of 0.5 nM. Primer sequences are listed in Table S6. Primer specificity was confirmed using melt-curve analysis. Bio-Rad CFX Manager 3.1 was used to calculate cycle threshold values using baseline subtracted curve fit and an auto-calculated single threshold. Final results were calculated using the ΔΔCt method with *CTCF* expression as the control. At least two replicates were analyzed per 24-well plate and at least two 24-well plates were analyzed per experiment.

## Relative importance of predictors

Expression data and predictors were scaled to z-scores across all sites in R using the scale function with default parameters before analyzing Pearson correlation. To calculate relative importance, a regression for gene expression changes by dCas9-VP16(10x) and dCas9-p300(core) were fit based on eight possible predictors using the lm function in R. Then, using the relaimpo package (Groemping, 2006), the relative importance of each predictor was calculated using the method described by Lindeman, Merenda, and Gold (Lindeman et al, 1980). Importance was normalized to sum to 1 for comparison. Distances to the nearest enhancers were calculated using H3K27ac peaks which do not overlap the TSS, as a proxy, using bedtools closest (Quinlan & Hall, 2010).

## Thermodynamic modelling

We used a modified version of the statistical thermodynamic model implemented in Gertz et al (2009), originating from the Shea–Ackers formalism (Shea & Ackers, 1985).

For a system state s, the relative energy of that state is:

$$E_s = \sum_{i=1}^{N} \left[ \sigma_i \times q_i + \sum_{j=i+1}^{N} \left( \sigma_{i,j} \times \omega_{i,j} \right) \right]$$

where $N$ is the number of ERBSs plus 1 for the promoter, $\sigma$ is a binary variable that denotes whether an ERBS or promoter is bound (0 for

unbound and 1 for bound), q values represent protein:DNA interactions, and ω values represent protein:protein interactions. Whereas relative q values are often fit using position weight matrices, in this case, gRNAs determine binding and we, therefore, leave the q values as free parameters.

We then calculate a thermodynamic weight for each state s as:

$$W_s = e^{-E_s/RT}$$

where $R$ is the gas constant and $T$ is temperature set at 37°C.

We define an experimental state, conditional on which enhancers are being targeted, which can be thought of as the union of possible states. For example, if sites 1 and 2 are being targeted, possible system states include site 1 is bound, site 2 is bound, both are bound, or none are bound; however, states with site 3 bound are considered highly unlikely and not considered.

Therefore, the probability of RNAPII being in a certain experimental state e is given by the partition:

$$p(RNAPII)_e = \frac{\sum_{s=1}^{2^N} W_s \times \delta(RNAPII) \times \delta(exper)}{\sum_{s=1}^{2^N} W_s \times \delta(exper)}$$

where $\delta(RNAPII)$ is a $\delta$ function which is equal to 1 when $RNAPII$ is bound in a given system state and 0 otherwise. $\delta(exper)$ is a $\delta$ function corresponding to whether a system state is possible, given which regulatory regions are being targeted in the experimental state. $2^N$ is the number of possible experimental states given $N$ regulatory regions.

Gene expression was assumed to be correlated with the probability of RNAPII being bound to the promoter. When fitting our parameters, we maximized the value of the following correlation:

$$cor[Expression, P(RNAPII)]$$

where $cor$ is the Pearson correlation across all experimental states tested.

Random starts were chosen as a set of 10 random numbers between the limits of −5 and 5 using the runif function in R. To control for potential bias resulting from the optimization algorithm used to fit the parameters, the parameters were fit using both the limited memory Broyden–Fletcher–Goldfarb–Shannon with box constraints (L-BFGS-B) algorithm (Byrd et al, 1995), which is a gradient descent-based method and the Hooke and Jeeves Pattern Search Optimization method (Hooke & Jeeves, 1961) which does not rely on gradient descent. An interface to these algorithms was implemented in R by John C Nash in the optimr package (Nash, 2016). Using both algorithms, 1,000 iterations were run, resulting in 2,000 parameter fits. We then selected parameter sets which correlated with gene expression data reasonably well (within 0.1 of an optimal correlation coefficient) for downstream analysis and plotting.

Inhibition by SID(4x)-dCas9-KRAB was modeled in the same way with the exception that sites not targeted were defined as "active." We, therefore, assume that if a site is targeted with SID(4x)-dCas9-KRAB, it is completely inactive. The model is again fit on the correlation between RNAPII occupancy and previously measured qRT-PCR data (Carleton et al, 2017).

## Data Availability

The ChIP-seq data are available at the Gene Expression Omnibus under accession GSE133300.

## Supplementary Information

## Acknowledgements

This work was supported by National Institute of Health (NIH)/National Human Genome Research Institute (NHGRI) R01 HG008974 to J Gertz and the Huntsman Cancer Institute. Research reported in this publication utilized the High-Throughput Genomics Shared Resource at the University of Utah and was supported by NIH/National Cancer Institute (NCI) award P30 CA042014. We thank Jeffery Vahrenkamp for analysis advice and we thank K-T Varley as well as Gertz and Varley lab members for their valuable comments on the research and the manuscript.

### Author Contributions

M Ginley-Hidinger: conceptualization, investigation, methodology, and writing—original draft, review, and editing.
JB Carleton: investigation, methodology, and writing—review and editing.
AC Rodriguez: investigation and writing—review and editing.
KC Berrett: investigation and writing—review and editing.
J Gertz: conceptualization, funding acquisition, methodology, and writing—original draft, review, and editing.

### Conflict of Interest Statement

The authors declare that they have no conflict of interest.

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
