## [Reviewer comments · Life Science Alliance]

Life Science Alliance

Sufficiency analysis of estrogen responsive enhancers using synthetic activators

Matthew Ginley-Hidinger, Julia Carleton, Adriana Rodriguez, Kristofer Berrett, and Jay Gertz
DOI: <https://doi.org/10.26508/lsa.201900497>

Corresponding author(s): Jay Gertz, Huntsman Cancer Institute, University of Utah and Matthew Ginley-Hidinger, University of Utah

Review Timeline:	Submission Date:	2019-07-22
	Editorial Decision:	2019-07-22
	Revision Received:	2019-07-23
	Editorial Decision:	2019-09-11
	Revision Received:	2019-09-17
	Accepted:	2019-09-18

Scientific Editor: Andrea Leibfried

Transaction Report:

Please note that the manuscript was previously reviewed at another journal and the reports were taken into account in the decision-making process at Life Science Alliance.

Reviewer #1 Review

Report for Author:

This paper follows up a study describing the dissection of estrogen receptor binding site (ERBS) contribution to gene expression via Enhancer-i, a multiplex CRISPR interference that uses a SID-Cas9-KRAB. Enhancer-i allowed measurement of combinatorial relationship between ERBS, highlighting synergistic effect and distance to TSS influence on enhancer strength.

In this paper, Ginley-Hidinger et al. study the same ERBS surrounding the 3 genes previously studied (plus a new one) with a new approach, CRISPR activation (CRISPRa). As opposed to Enhancer-i that test the necessity of a genomic region for regulatory activity, with CRISPRa, the authors aim to test the sufficiency of an enhancer to affect the transcription of its associated gene.

Two different constructs are tested for their ability to activate gene expression when targeted to ERBSs: dCas9-VP16(10x) and dCas9-p300. dCas9-VP16(10x) showed activation of gene expression to a higher level than dCas9-p300 on the targeted ERBS. dCas9-VP16(10x) is further on used to target individual and combinations of ERBS regulating the same gene. The results show variable efficiency of the CRISPRa to induce transcription and in some cases cumulative effect when 2 or 3 ERBS are targeted simultaneously but no synergistic effect is measured as opposed to what has been previously observed with Enhancer-i. From those results, a thermodynamic modelling of RNA Pol II recruitment by ERBS is then realized that confirms the independence of the ERBS in gene activation. Enhancer-a results are also compared with the Enhancer-i data.

The authors conclude that the synergistic behaviour of ERBS is due to how ER and its cofactors are recruited and is not observed with CRISPRa where the cofactors are directly recruited.

The approach to use CRISPRa as a mean to test how multiple activators at different locations influence transcriptional activation is interesting. However, this assay is limited in the sense that it cannot provide insight into the innate capability of an enhancer to affect transcription, since this innate function is over-ruled by the CRISPR-dCas9-VP, as through guide RNA binding, the activator protein directly recruits RNA polII.

The analysis of multiple simultaneous activators on different locations could be interesting to test for example the effect of repressors, or the influence of chromatin looping. However, such analyses were not performed. We find this overall an interesting study, although the technique has limited value to understand "enhancer function"

Comments:

- The validation of the CRISPRa tool has been done very thoroughly to show that it is indeed responsible for the effects observed. There is however no individual validation of the targeting efficiency of the guides. The guide-to-enhancer efficiency can differ, the distance to the TSS differs, the distance to each other and to TAD border differ, etc. The authors do not take these aspects into account for the interpretation of the differences in activation. For example, the authors write "Native activation potential unique to the site"- I do not understand what in the site can affect the function of the synthetic activator, since all normal function is overruled by the CRISPR-dCas9 interaction. The main factor here seems to be distance to TSS, the location of the enhancer relative to other enhancers, and the location of the enhancer relative to TAD borders. Can the authors relate the tested enhancers with distance to TSS and relation to TAD borders? Also, the comparison of gene activation directly driven from the TSS (as mostly performed for CRISPR-a) would have been better included as control.
- If other, non-ERBS, DHSs would be targeted (they were avoided in the design), would they also work? One could reason that they should work since the activator is brought in proximity to the TSS. What if non-DHS sites are targeted? Why can non-DHS sites not be used to activate the target? Is binding of gRNA dependent on nucleosome positioning and open chromatin?
- The H3K27Ac profiling after dCas9-VP16 recruitment is interesting; it shows H3K27Ac signal (fig 2C), also at the other enhancers in the same locus. This illustrates, as expected, that multiple enhancers form a chromatin hub (see also work by Ana Pombo on nuclei slicing). What is the impact of such chromatin hubs on the Enhancer-a assay when multiple guides are used?
- In theory, if the DNA sequence of an enhancer would be shuffled/randomized, and placed instead of the original sequence, then targeted by a guide RNA targeting the shuffled sequence, do the authors expect the same or a different outcome of that site? Can this thought experiment be discussed in the text? If the authors expect a different outcome, then that has to be substantiated by experiments and analyses, and discussed in the text (how does the local enhancer context influence the gRNA-dCas9-VP16 function).
- Overall this technique does not necessarily bring new insight regarding how ERBS regulates transcriptional activity; the assay is not well suited to unravel a particular transcriptional program, but rather to study general effects of enhancer-to-target distance. This is not described clearly enough. Perhaps the authors can explain this better in the text, to avoid confusion and over-interpretation.
- Figure 4 - TD model: in other TD models of gene regulation, the TF-binding site strength (q_1 , q_2 , ...) is determined by the PWM scores. With guide RNAs, this interaction seems to be more difficult to model/parametrize. Can this be discussed?

Reviewer #2 Review

Report for Author:

Summary

This paper is based on a previous paper from the same group where they use 3 out of 4 genes and check the collaborative control of gene expression by TFBSs in close proximity. In the current manuscript they used already published dCas9 activator complexes (dCas9-VP16 and dCas9-p300) to replace the ER signaling by these activators. The authors found that dCas9-p300 is not working as well as dCas9-VP16. While the targeted sites have a clear effect on gene expression, which was bigger than the effect of targeting non-ER sites, there was no marked difference in terms of H3K27ac signal, which accumulated at both sites. This let the authors conclude that H3K27ac is not a very good predictor of gene expression. They see a saturation effect on gene expression when activating single enhancers using synthetic activators, which is in contrast to the synergistic behavior observed in the repression model of their previous work. They suggest that the synergy between ERBS likely occurs in cis, where the recruitment of ER and its cofactors, such as HATs (Hanstein et al., 1996; Shang et al., 2000), is synergistic between ERBS. However, if the cofactors are directly recruited to the ERBS, as is the case with CRISPRa, then synergy no longer occurs.

The overall findings of the paper are highly relevant and add significant insights into the current discussion about how enhancers activate gene expression. Unfortunately only at a very small set of enhancers, so general conclusions have to be taken with a grain of salt. Also the conclusions remain somewhat unsatisfying since it is not clear whether the lack of synergy is due to the fact that VP16(10) is such a strong activator, already preloaded with mediator and other activating TFs that is strong enough on its own, whereas in a typical ER binding event, the synergy would arise from interaction of the sites in terms of recruiting the activators. Thus our question is whether the authors can leverage the p300 data (e.g. also add it to the model) to draw some conclusions about this.

Summary of major points to be addressed:

- 1) Statement about correlation of relative expression and ER induction too strong (S1C) this is only based on 4 points and HES2 and MMP17 don't agree in terms of relative strength.
- 2) Fig 1B: why does HES2 have no adjacent regions? Why is it selected at all? More information should be given in manuscript to clarify by which criteria respective TFBS and adjacent regions were chosen. Are there any other regulatory regions (TFBS) of other TFs, not ER, overlapping with the ER TFBS or adjacent regions? Why were TFBS in the gene bodies chosen as adjacent regions - one would expect quite different properties of them? Are these gene body regions even overlapping with exons? Also it remains unclear why the authors did not test any adjacent regions for the HES2 gene.
- 3) Fig 1C: is there any explanation of why P300 has an effect in 2 out of the 4 genes? This might be

worth going a bit more into detail since it could shed light on whether just acetylation (and no activator recruitment as with VP16(10)) would act more synergistic. It was not clear why the model was not tried to explain the p300 data. The authors showed that synthetic activators at the respective TFBS increase gene expression as compared to the activation of the adjacent regions, however more explanation is needed of why dCas9-p300 worked only for 2 cases out of 4. Taking into account that, as discussed in the discussion, the VP16 construct can already be preloaded with the Mediator complex, it would be interesting to follow up with the p300 to see the unbiased gene regulation scheme, also in terms of the hypothesis of synergy vs independent activation.

4) Fig2A: same plot for P300 is missing

5) Fig 2A: are there replicates? To check the binding of dCas9 constructs ChIP-seq was performed, however there is no information on replicates. Would be good to show

6) Lack of correlation (S3D) between H3K27ac and gene expression This lack of the correlation between H3K27ac and gene expression is quite striking. However, this conclusion is based on a complicated normalization procedure based on a small number of 'control genes' thus opening the possibility that the lack of correlation is based on artifacts arising from the normalization procedure. More quality control plots and intermediate numbers are required to strengthen the conclusion (e.g. to see unnormalized values and ratios to exclude the effect of dividing by the small numbers). It is also concerning that for the binding site 1 of the HES2 gene has a peak of H3K27ac even when targeting different TFBS from that gene.

7) Fig 1A: Why are these regions chosen? Explain more

8) Fig S2E: HES2 site 1 is always on what is the explanation for this? Since this is a very crucial site for the later conclusions it is important to understand why it is so different

9) Fig4: why 2 algorithms? Why 500 parameter sets? Show the individual algorithms. Show the final convergence of the fit and the final correlation. The authors developed a thermodynamic model predicting the RNAPII binding (should correlate strongly with gene expression) to identify which interactions are crucial in the system. While this is an elegant model to explain the data, it should be described in a bit more details. For example, it is not clear why only best performed 500 parameter sets were chosen, why they have combined the results from 2 algorithms, what the bimodal distributions of most of the parameters mean.

10) Global conclusion: is it the property of these enhancer regions that makes them work independently, or is it VP16? Can you do the model based on p300? Would that give insights into the mechanism at a different level (excluding the strong biases from the transactivator)

Minor Points:

1) The introduction should be phrased more clearly. The necessity vs sufficiency concept could be explained more clearly (and earlier in the intro)

2) S2D-F: show axis scales in the figure

Reviewer #3

Report for Author:

Ginley-Hidinger et al. aimed to test the sufficiency of estrogen-responsive enhancers by using synthetic dCas9-based activators (CRISPRa) that are targeted to 12 previously identified estrogen receptor binding sites (ERBSs) linked to 4 genes. This study essentially builds on previous work by the same group in which a CRISPR-based enhancer interference ("enhancer-i") approach was used, involving the coupling of dCas9 to a repressive (rather than an activating) domain such as KRAB, to target many of the same enhancers as probed in the current paper.

To the authors' credit, they acknowledge that the outcome of these two studies is fundamentally different in that enhancer-i uncovered clear synergistic enhancer behavior, whereas CRISPRa revealed mostly independent/additive behavior. Based on the provided data, this conclusion appears sound. However, what is then the major take home message of this study, beyond the rather technical observation that CRISPRa constructs, and especially the dCas9 one coupled to 10 (!) VP16 domains, are not useful to study enhancer function since they act as sledge hammer-like systems that override the subtler mechanisms of endogenous gene regulation? This is perhaps the most important issue of this study, since, as it stands, the offered conceptual advance is minor at best.

Next to this major issue, there are several other aspects of this study that are only superficially covered even though their more in-depth, mechanistic analysis would significantly improve the paper's impact:

* For example, to rationalize the differences between their two studies, the authors indicate that endogenous gene regulation is driven by a synergy between ERBSs, as mediated by the synergistic recruitment of ER and its co-factors, in contrast to CRISPRa, which directly recruits activating cofactors to ERBSs. The authors write this as a factual statement, but no formal molecular evidence to support this statement and to more specifically demonstrate how these regulatory differences between the two utilized systems are molecularly encoded is provided. In other words, a big, question that this study does not resolve is why do ERBSs cooperate when targeted by ERa, but not when targeted by dCas9-VP16?

*The authors observe that dCas9-p300 activates genes to a lower level compared to dCas9-VP16, indicating that VP16 is likely contributing to gene activation "in other ways" (beyond mere H3K27Ac deposition) than p300. Here, the authors have an opportunity to go to the regulatory heart of co-factor function, yet they settle unfortunately at the observational rather than the mechanistic level.

*The presented data are seemingly at odds with the authors' statement that H3K27Ac deposition by p300 and VP16 is similar. Indeed, as shown in Fig. 2C-D and Fig. S2D-E, the overall enrichment appears systematically lower for p300 compared to VP16 tracks. Fig. 2B summarizes the H3K27Ac data, showing that overall enrichment levels are comparable, but then how can the authors explain the provided track enrichment data, which clearly suggests otherwise?

*Does dCas9-p300 bind ERBSs to the same extent as dCas9-VP16? No dCas9-p300 ChIP-seq data is included in the manuscript so this question cannot be addressed based on the provided data. Also, why is the dCas9-VP16 ChIP data so variable and does this reflect technical or biological variation? If the latter, could this also influence the enhancer activation read-out?

*Finally, the authors show that sites that are adjacent to ERBSs but that are not targeted by ERa itself can also be targeted by CRISPRa, resulting in H3K27Ac deposition but not gene activation. Is

this observation at odds with the statement that CRISPRa-targeted enhancers operate independently? In other words, an important and equally unresolved question is why these adjacent sites are unable to act as independent enhancers, even though they seem to be activated themselves based on H3K27Ac. Do the authors detect eRNAs for these sites and does their failure to participate in the gene activation process reflect local chromosome conformation? Answering these questions from a mechanistic rather than observational point of view would again be important to increase the conceptual value of the current study.

July 22, 2019

Re: Life Science Alliance manuscript #LSA-2019-00497-T

Dr Jay Gertz
Huntsman Cancer Institute, University of Utah
Department of Oncological Sciences
2000 Circle of Hope
Salt Lake City, Utah 84112

Dear Dr. Gertz,

Thank you for transferring your manuscript entitled "Sufficiency analysis of estrogen responsive enhancers using synthetic activators" to Life Science Alliance. The manuscript was assessed by expert reviewers at another journal before, and the editors transferred those reports to us with your permission.

The reviewers who evaluated your work had split views on the definitive value provided. Reviewer #2 appreciated your work, while the other reviewers were more critical. Importantly, the reviewers thought that the VP16 system is not ideal for studying endogenous enhancer behaviour as VP16 strength may overrule all inherent features of native enhancers. They further noted some overstatements. The reviewers also provided constructive input on how to address - at least partially - their concerns.

Based on the reports already at hand, and as discussed with you already pre-submission, we would like to invite you to submit a revised version of your work for publication in Life Science Alliance. We would expect a point-by-point response to all concerns raised and accordingly changes in the manuscript text. At the experimental level, the first point of reviewer #1 should get addressed, the p300 data extended (reviewer #2 general statement; reviewer #3 point 2), and points 1, 2, 3 and 6 of reviewer #2 should get addressed. Points 3 and 4 of reviewer #3 should get addressed, too. A full mechanistic understanding (reviewer #3) will not be required for publication here, we think that extending the p300 data will address this criticism sufficiently.

Please note that papers are generally considered through only one revision cycle, so strong support from the referees on the revised version is needed for acceptance.

Thank you for this interesting contribution to Life Science Alliance. We are looking forward to receiving your revised manuscript.

Sincerely,

Andrea Leibfried, PhD

Executive Editor
Life Science Alliance
Meyerhofstr. 1
69117 Heidelberg, Germany
t +49 6221 8891 502
e a.leibfried@life-science-alliance.org
www.life-science-alliance.org

B. MANUSCRIPT ORGANIZATION AND FORMATTING:

We thank the reviewers for their comments. Our responses are shown in red below.

Reviewer #1:

Comments:

The validation of the CRISPRa tool has been done very thoroughly to show that it is indeed responsible for the effects observed. There is however no individual validation of the targeting efficiency of the guides. The guide-to-enhancer efficiency can differ, the distance to the TSS differs, the distance to each other and to TAD border differ, etc. The authors do not take these aspects into account for the interpretation of the differences in activation.

We thank the reviewer for pointing out the thorough testing of CRISPRa being responsible for the expression effects that we observe. We have now validated the guide RNA targeting efficiency of both dCas9-VP16(10x) and dCas9-P300(core) using ChIP-seq with an antibody that recognizes the HA tag on dCas9 in duplicate (Figure 2A, see below). The targeting efficiency of both fusions is similar and the amount of ChIP-seq signal at individual sites does not correlate with the ability of a site to activate when bound by the synthetic activator (Figure S3C, see below). These results are now mentioned in the results section:

To test whether CRISPRa is successfully targeted to the intended genomic regions surrounding our genes of interest, we conducted a Chromatin Immunoprecipitation followed by sequencing (ChIP-seq) experiment using an antibody that recognizes an HA epitope tag on dCas9. dCas9-VP16(10x) was targeted to 19 loci, consisting of 12 ERBS, 6 ERBS-adjacent regions and the *IL1RN* promoter (Figure 1B). At all of these loci, we observed a distinct HA (dCas9) signal at the targeted site when compared to non-targeted controls (Figures 2A,B and S2A-C). Successful targeting of dCas9-VP16(10x) to ERBS-adjacent regions indicates the lack of activation from these regions is based on genomic properties of the adjacent regions and not a result of the targeting efficiency. dCas9-p300(core) was successfully targeted to all 12 ERBS (Figure 2A).

...

However, we found no significant correlation between expression activation and fold change HA ChIP-seq signal, fold change H3K27ac ChIP-seq signal or fold change RNAPII ChIP-seq signal (Figure S3C-E, see below).

For example, the authors write "Native activation potential unique to the site"- I do not understand what in the site can affect the function of the synthetic activator, since all normal function is overruled by the CRISPR-dCas9 interaction. The main factor here seems to be distance to TSS, the location of the enhancer relative to other enhancers, and the location of the enhancer relative to TAD borders. Can the authors relate the tested enhancers with distance to TSS and relation to TAD borders?

We agree that the question of what gives an enhancer activation potential is very interesting. We have now included an additional correlative analysis to determine which factors may predict gene activation levels. In this analysis we analyzed a variety of variables which may influence fold change gene expression including targeting efficiency (HA ChIP-seq), base and fold change H3K27ac, base and fold change RNAPII occupancy, distance to the TSS, distance to nearest other enhancer and number of other transcription factors bound at that site (out of 19 analyzed TFs). All sites tested, with the exception of CISH-3 are within the same TAD boundary as the gene. We found that the number of other TFs was the highest correlate for VP16(10x) activation while base RNAPII levels was the highest correlate for p300 activation (Figure 5D). To analyze the relative importance of these different variables, we used a regression-based analysis using the "lmg" method from the relaimpo package in R where combinations of predictors are used to assess the impact of a single predictor on the percent variance explained by the model. This confirmed that the best predictor of VP16(10x)-mediated activation was presence of other transcription factors while the best predictor for p300(core) was base RNAPII levels. This analysis is now included in figure 5 (below) in the paper and is mentioned in the results, discussion and methods:

Results: To determine what genomic traits are predictive of enhancer strength, we looked at a set of 8 possible predictors of activation by both dCas9-p300(core) and dCas9-VP16(10x). The strongest correlated variables to dCas9-VP16(10x) and dCas9-p300(core) activation included the number of transcription factors (TFs) present at each site, the base level of RNAPII ChIP-seq signal and the base level of H3K27ac ChIP-seq signal (Figures 5D, S5E-H). Using a multiple regression-based method, the relative importance of each variable was calculated (see methods) (Groemping, 2006). This revealed that the best predictor of dCas9-VP16(10x) activation was the number of TFs present at the site, while the best predictor for dCas9-p300(core) mediated activation was the base amount of RNAPII present at each site (Figure 5E). The number of TFs bound to each site was determined using publicly available ChIP-seq data (ENCODE Project Consortium, 2012) for 19 different TFs in Ishikawa cells and looking for transcription factors that were bound to the 12 enhancers of interest as well as

adjacent sites (Tables S2 and S3). While no TF was bound solely to strongly activated sites, we found certain TFs, such as TCF12 and ZBTB7A, were bound more often to enhancers that exhibited strong activation when targeted with synthetic activators (Table S2). Overall, these data suggest that dCas9-VP16(10x) and dCas9-p300(core) may have different requirements for target gene activation.

Discussion: Analysis of potential genomic factors which may determine activation potential revealed that the best predictor of dCas9-VP16(10x) activation is the amount of other transcription factors present at the site. A battery of other transcription factors may help stabilize dCas9-VP16(10x) or may perform orthogonal roles in enhancer maturation. This finding indicates that the underlying DNA sequence may be important for regulation by synthetic activators as they may be working together with other TFs. For dCas9-p300(core), the most important predictor is the base amount of RNAPII already present at the site, which could point to an important difference between the synthetic activators in their ability to recruit RNAPII, as VP16 has been shown to recruit basal transcription factors (Hirai et al., 2010). The unique predictors from this analysis suggests that different modes of activation are used by dCas9-VP16(10x) and dCas9-p300(core), which warrants further study.

Methods:

Relative importance of predictors

Expression data and predictors were scaled to z-scores across all sites in R using the scale function with default parameters before analyzing Pearson correlation. To calculate relative importance, a regression for gene expression changes by dCas9-VP16(10x) and dCas9-p300(core) were fit based on 8 possible predictors using the lm function in R. Then, using the relaimpo package (Groemping, 2006), the relative importance of each predictor was calculated using the method described by Lindeman, Merenda and Gold (LMG) (Lindeman et al., 1980). Importance was normalized to sum to 1 for comparison. Distances to the nearest enhancers were calculated using H3K27ac peaks which do not overlap the TSS, as a proxy, using bedtools closest (Quinlan and Hall, 2010).

Also, the comparison of gene activation directly driven from the TSS (as mostly performed for CRISPR-a) would have been better included as control.

We agree that a comparison to promoter targeting is a good experiment. We have done this analysis and found that targeting enhancers in combination is often equal to or exceeds targeting the promoter in activation (Figure 1C-F shown below). We have included the following text in our results: For comparison, we also targeted our constructs to the promoter regions of these genes and found that targeting ERBS often results in equal or greater activation than targeting the promoter (Figure 1C-F).

If other, non-ERBS, DHSs would be targeted (they were avoided in the design), would they also work? One could reason that they should work since the activator is brought in proximity to the TSS. What if non-DHS sites are targeted? Why can non-DHS sites not be used to activate the target? Is binding of gRNA dependent on nucleosome positioning and open chromatin?

While we think this is an interesting question, we believe that it is tangential to the main focus of this manuscript. In addition, dCas9-P300 has been targeted to several DHS sites (Klann et al Nature Biotech 2017) in a high throughput manner and some activate while others do not. Therefore, simply being a DHS site is not sufficient for CRISPRa activation. For the non-DHS sites, we targeted six of these regions (adjacent sites) and none of them were able impact gene expression when targeted with CRISPRa, even though these sites are well targeted by the fusion and gain H3K27ac (Figure 2B,D shown below). Therefore, it is not simply the recruitment of an activator to within a certain distance of the TSS.

The H3K27Ac profiling after dCas9-VP16 recruitment is interesting; it shows H3K27Ac signal (fig 2C), also at the other enhancers in the same locus. This illustrates, as expected, that multiple enhancers form a chromatin hub (see also work by Ana Pombo on nuclei slicing). What is the impact of such chromatin hubs on the Enhancer-a assay when multiple guides are used?

We agree that this is an interesting observation and suggests that the enhancers could be in close 3D proximity with one another. To determine if some enhancers are in close spatial proximity, we performed HiC on these cells without CRISPRa. Unfortunately, the resolution was not high enough to detect significant interactions between any of the queried enhancers.

In theory, if the DNA sequence of an enhancer would be shuffled/randomized, and placed instead of the original sequence, then targeted by a guide RNA targeting the shuffled sequence, do the authors expect the same or a different outcome of that site? Can this thought experiment be discussed in the text? If the authors expect a different outcome, then that has to be substantiated by experiments and analyses, and discussed in the text (how does the local enhancer context influence the gRNA-dCas9-VP16 function).

This is an interesting idea/question. Now that we have discovered that sites with many TFs bound are more likely to activate gene expression when targeted by synthetic activators, we believe that the underlying DNA sequence matters. Unfortunately, the time and effort required to shuffle enhancer sequences and target them makes the experiment more suitable for another study. We have now added this idea to the discussion:

Analysis of potential genomic factors which may determine activation potential revealed that the best predictor of dCas9-VP16(10x) activation is the amount of other transcription factors present at the site. A battery of other transcription factors may help stabilize dCas9-VP16(10x) or may perform orthogonal roles in enhancer maturation. This finding indicates that the underlying DNA sequence may be important for regulation by synthetic activators as they may be working together with other TFs.

Overall this technique does not necessarily bring new insight regarding how ERBS regulates transcriptional activity; the assay is not well suited to unravel a particular transcriptional program, but rather to study general effects of enhancer-to-target distance. This is not described clearly enough. Perhaps the authors can explain this better in the text, to avoid confusion and over-interpretation.

We agree that the CRISPRa system is not the perfect method for unraveling enhancer function, especially with respect to a specific transcription program, due to the potential recruitment of cofactors that are not involved in a particular enhancers function or the potential interference of transcription factor binding. However, we do feel that using estrogen signaling as an inducible model system, we were able to compare the recreation of an estrogen response to the interference of an estrogen response, which provided insight into how important each individual site is and how they work together. We believe that our results have important implications for both the technologies being used as well as the manner in which ER bound sites work together. These concepts are now included in the discussion:

There are important caveats to consider when using these synthetic activators including the possible recruitment of cofactors that do not normally bind to a particular enhancer or potential interference with TF binding by dCas9. However, we believe that the comparison between enhancer activation and enhancer interference has shed light on the consistent importance of individual sites as well as a key difference in how enhancers work together when bound by different transcriptional activators.

Figure 4 – TD model: in other TD models of gene regulation, the TF-binding site strength (q_1, q_2, \dots) is determined by the PWM scores. With guide RNAs, this interaction seems to be more difficult to model/parametrize. Can this be discussed?

The reviewer is correct that most TD models use PWMs to estimate relative q values. Since we don't have the equivalent for guide RNAs, we currently leave this as a free parameter that is fit by the model. This is now clarified in the methods:

While relative q values are often fit using position weight matrices, in this case guide RNAs determine binding and we therefore leave the q values as free parameters.

Reviewer #2:

The overall findings of the paper are highly relevant and add significant insights into the current discussion about how enhancers activate gene expression. Unfortunately only at a very small set of enhancers, so general conclusions have to be taken with a grain of salt. Also the conclusions remain somewhat unsatisfying since it is not clear whether the lack of synergy is due to the fact that VP16(10) is such a strong activator, already preloaded with mediator and other activating TFs that is strong enough on its own, whereas in a typical ER binding event, the synergy would arise from interaction of the sites in terms of recruiting the activators. Thus our question is whether the authors can leverage the p300 data (e.g. also add it to the model) to draw some conclusions about this.

We agree that the analysis of more sites would establish the generality of the conclusions; however, these experiments are fairly involved and we believe this is the largest such study to date that doesn't involve regulatory element screening. dCas9-VP16(10x) is likely a strong activator, but it can't be too strong, since estrogen activates gene expression to higher or similar levels at each gene even when dCas9-VP16(10x) is targeted to promoters (see new Figure 1), suggesting that VP16(10x) is not a terribly heavy hammer. In addition, E2 treatment causes larger inductions in H3K27ac and targeting dCas9-VP16(10x) does not cause activation when targeted to any nearby region, indicating that the strength of activation can be limited by the site targeted. These points have been added to the discussion:

One explanation for the independence between enhancers when bound by synthetic activators is that the synthetic activators are so strong that they override more subtle regulatory events that require synergy. However, we do not believe this to be the case, since an E2 induction leads to higher expression levels than either activator, especially dCas9-p300(core).

We thank the reviewer for the suggestion of targeting individual and combinations of sites with the dCas9-P300(core) fusion and modeling the data. We agree that using a weaker activator has the potential to uncover synergy that is masked by a stronger activator. We have done this experiment for the genes that exhibit significant induction when all sites are targeted (*CISH*, *MMP17* and *HES2*). The results indicate that when these sites are targeted by dCas9-p300(core) they work independently to regulate gene expression. The individual targeting results and the dCas9-p300(core) thermodynamic modeling results have been added to Supplementary Figure 4 (see below). We have included the following in the results section in two places:

Targeting dCas9-p300(core) to individual ERBS resulted in a lower, but correlated, activation than dCas9-vp16(10x) ($r = 0.633$, Figure S4A-D), indicating that the relative strength of enhancers may be independent of the synthetic activator used, while absolute strength can be controlled by the strength of the synthetic activator.

...

It is possible that the observed independence of synthetically activated ERBS is due to the strength of dCas9-VP16(10x), which may override the subtleties of enhancer synergy. We therefore used thermodynamic models to analyze activation by a weaker activator, dCas9-p300(core), at the 3 genes which can be activated by this construct. Again, we observed an independent relationship at *MMP17* (Figure S4E). At *CISH* and *HES2*, we observed bimodal distributions of some interaction parameters, where a subset of fitted models indicate synergy between enhancers; however, the majority of model fits indicate independence between enhancers when bound by dCas9-p300(core) (Figure S4F,G). The presence of multiple local optima might be due to low levels of activation and lower signal-to-noise ratios. Overall, we see that sites activated by dCas9-p300(core) do not display strong patterns of cooperativity, consistent with the results from dCas9-VP16(10x) targeting.

Summary of major points to be addressed:

1) Statement about correlation of relative expression and ER induction too strong (S1C) this is only based on 4 points and HES2 and MMP17 don't agree in terms of relative strength.

We apologize for the strong language and have toned it down. The following language is now present in the results:

The level of gene expression driven by dCas9-VP16(10x) was somewhat correlated with the fold change in gene expression seen with a 17 β -estradiol (E2) induction at this set of four genes ($r = 0.8$) (Figure S1C).

2) Fig 1B: why does HES2 have no adjacent regions? Why is it selected at all? More information should be given in manuscript to clarify by which criteria respective TFBS and adjacent regions were chosen. Why were TFBS in the gene bodies chosen as adjacent regions - one would expect quite different properties of them? Are these gene body regions even overlapping with exons? Also it remains unclear why the authors did not test any adjacent regions for the HES2 gene.

We originally chose adjacent regions nearby FHL2, MMP17, and CISH ER binding sites, because they were involved in our original Cell Systems study (Carleton et al 2017). We then added HES2 and its ER binding sites as another example of a gene with multiple ER bound sites in the vicinity. We have now added additional information about why we chose the adjacent regions in the results:

To determine if the activation potential is specific to ERBS, we targeted dCas9-VP16(10x) and dCas9-p300(core) to a total of 6 regions surrounding *MMP17*, *CISH* and *FHL2* that are at most 8 kb away from ERBS discussed above (or the TSS in the case of FHL2-B) (Figure 1B). Regions with low DNase I hypersensitivity signal (Gertz et al., 2013) were chosen, in order to limit the probability that the locus was an active regulatory region controlled by other transcription factors. As *HES2* is in a highly active region with several DNase I hypersensitive sites and histone H3 lysine 27 acetylation positive loci, multiple nearby genes and many transcription factor binding events, we were unable to choose sites that were not potential regulatory regions at this locus. In choosing adjacent regions, we aimed to keep the distance to the TSS similar without being too close to the ER bound site. We observed some transcription factors binding to the chosen adjacent regions, notably at CISH-A (Table S3). When targeting the ERBS adjacent sites, we did not observe significant activation over the control of targeting the *IL1RN* promoter (Figure 1C-E). The inability of ERBS adjacent regions to regulate gene expression with synthetic activators indicates specificity when testing sufficiency of regulatory regions with CRISPRa and differences in activation potential between ERBS and nearby non-ERBS.

In the case of adjacent site FHL2-B, our design would have put the region at the TSS, so we moved it into the gene body. It does not overlap with any exons. We agree that targeting regions inside of genes may have different results; however, we do not see activation at any of the three genes tested.

Are there any other regulatory regions (TFBS) of other TFs, not ER, overlapping with the ER TFBS or adjacent regions?

We agree that a characterization of other TFs that bind to these sites is important information, that also led to an interesting conclusion about the ability of dCas9-VP16(10x) to activate expression (see our response to reviewer 1's 2nd comment above). Information about TFs that overlap with each region has been added as Tables S2 and S3 and discussed in the results:

The number of TFs bound to each site was determined using publicly available ChIP-seq data for 19 different TFs in Ishikawa cells (ENCODE Project Consortium, 2012) and looking for transcription factors that were bound to the 12 enhancers of interest as well as adjacent sites (Tables S2 and S3). While no TF was bound solely to strongly activated sites, we found certain TFs, such as TCF12 and ZBTB7A, were bound more often to enhancers that exhibited strong activation when targeted with synthetic activators (Table S2).

Table S2. Transcription factor binding at ERBS. 1 = bound, 0 = not bound

	MMP17- 1	MMP17- 2	MMP17- 3	CISH- 1	CISH- 2	CISH- 3	FHL2- 1	FHL2- 2	FHL2- 3	HES2- 1	HES2- 2	HES2- 3
GR	0	0	0	0	0	0	0	0	0	0	0	0
RAD21	0	0	0	0	0	0	0	0	0	1	1	0
CEBPB (SC150)	0	0	1	0	1	0	0	0	1	1	0	0
USF1	0	1	0	0	0	0	0	0	0	1	0	1
YY1 (SC281)	0	0	0	0	0	0	0	0	0	1	0	1
SRF	0	0	0	0	0	0	0	0	0	0	0	0
TAF1	0	0	0	0	0	0	0	0	0	0	0	0
TCF12	1	0	0	1	1	0	1	1	1	1	1	1
NFIC (SC81335)	1	0	1	0	1	0	1	1	1	1	1	1
FOXA1 (SC 6553)	0	0	0	0	0	0	0	0	0	0	0	0
MAX	1	0	0	1	1	0	0	1	0	1	0	1
NRSF	0	0	0	0	0	0	0	0	0	0	0	0
CREB1 (SC 240)	0	0	0	0	0	0	0	0	0	0	0	0
P300	1	0	0	0	1	0	1	1	1	1	1	1
POLII	0	0	0	0	1	0	0	0	0	1	1	1
TEAD4 (SC101184)	1	0	0	0	0	0	1	1	1	1	0	1
ZBTB7A	1	0	0	0	0	0	1	0	0	1	0	1
FOXM1 (SC 502)	0	0	0	0	0	0	0	0	1	1	0	0
TOTAL TFS	6	1	2	2	6	0	5	5	6	12	5	9

Table S3. Transcription factor binding at adjacent regions. 1 = bound, 0 = not bound

	MMP17-A	MMP17-B	CISH-A	CISH-B	FHL2-A	FHL2-B
GR	0	0	0	0	0	0
RAD21	0	0	1	0	0	0
CEBPB (SC150)	0	0	0	0	0	0
USF1	0	0	0	0	0	0
YY1 (SC281)	0	0	0	0	0	0
SRF	0	0	0	0	0	0
TAF1	0	0	1	0	0	0
TCF12	0	0	1	0	0	0
NFIC (SC81335)	0	0	1	1	0	0
FOXA1 (SC 6553)	0	0	0	0	0	0
MAX	0	0	1	0	0	0
NRSF	1	0	0	0	0	0
CREB1 (SC 240)	0	0	0	0	0	0
P300	0	0	1	0	0	0
POLII	0	0	1	0	0	0
TEAD4 (SC101184)	0	0	1	0	0	0
ZBTB7A	0	0	1	0	0	0
FOXM1 (SC 502)	0	0	0	0	0	0
TOTAL TFS	1	0	9	1	0	0

3) Fig 1C: is there any explanation of why P300 has an effect in 2 out of the 4 genes? This might be worth going a bit more into detail since it could shed light on whether just acetylation (and no activator recruitment as with VP16(10)) would act more synergistic. It was not clear why the model was not tried to explain the p300 data. The authors showed that synthetic activators at the respective TFBS increase gene expression as compared to the activation of the adjacent regions, however more explanation is needed of why dCas9-p300 worked only for 2 cases out of 4. Taking into account that, as discussed in the discussion, the VP16 construct can already be preloaded with the Mediator complex, it would be interesting to follow up with the p300 to see the unbiased gene regulation scheme, also in terms of the hypothesis of synergy vs independent activation.

In the original manuscript, we did not test dCas9-p300(core) at individual sites, since we didn't observe much activation. We have now done this experiment and built thermodynamic models as discussed in response to reviewer 2's general comment above. dCas9-p300(core) activation from individual sites and the related modelling results have been added to Figure S4. Notably, we now see activation from 3 of 4 genes with additional replicates added. We believe that *FHL2* has a small dynamic range, even for activation with dCas9-VP16(10x), so we are not able to detect changes from dCas9-p300(core). These data also let us explore the differences in activation between the two CRISPRa constructs and we identified different predictors of gene activation (see response to reviewer 1's 2nd comment above).

4) Fig2A: same plot for P300 is missing

5) Fig 2A: are there replicates? To check the binding of dCas9 constructs ChIP-seq was performed, however there is no information on replicates. Would be good to show

We have now performed the HA ChIP-seq experiment for dCas9-p300(core) and found similar targeting efficiencies between the two constructs (Figure 2A, see below). We have also performed these (and all ChIPs in duplicate in order to perform statistical analyses). dCas9-p300(core) and dCas9-vp16(10x) bind all targeted regions (please see our response to reviewer 1's 1st comment above).

6) Lack of correlation (S3D) between H3K27ac and gene expression This lack of the correlation between H3K27ac and gene expression is quite striking. However, this conclusion is based on a complicated normalization procedure based on a small number of 'control genes' thus opening the possibility that the lack of correlation is based on artifacts arising from the normalization procedure. More quality control plots and intermediate numbers are required to strengthen the conclusion (e.g.

to see unnormalized values and ratios to exclude the effect of dividing by the small numbers). It is also concerning that for the binding site 1 of the HES2 gene has a peak of H3K27ac even when targeting different TFBS from that gene.

We have now replicated the H3K27ac ChIP-seq experiments which has reinforced the conclusion that fold change H3K27ac is not predictive of activation. We changed the normalization procedure to use all overlapping peak regions for normalization in the hope that it is more robust, which is now mentioned in the methods section:

For H3K27ac ChIP-seq and RNAPII ChIP-seq, counts were then normalized to the average read depth within all overlapping peaks for a given antibody. H3K27ac levels before normalization show the same activation trends, with different baseline levels (Figure S2G).

For HES2, there is a peak of H3K27ac at site 1 even at control levels, indicating that this site has a baseline level of acetylation present in these cells. With the additional data and the new normalization, the correlation with gene expression is still non-existent (Figure S3D, see response to reviewer 1's 1st comment above). In order to be more comprehensive, we have also included non-normalized data, as reads per million, in Figure S2G (see below).

7) Fig 1A: Why are these regions chosen? Explain more

Please see our response to reviewer 2's 2nd comment above.

8) Fig S2E: HES2 site 1 is always on what is the explanation for this? Since this is a very crucial site for the later conclusions it is important to understand why it is so different

The reviewer brings up an interesting point. HES2 has a basal level of H3K27ac in controls that increases when targeted or upon E2 induction. In table S2 (see our response to reviewer 2's 2nd comment above), we see that 12 other transcription factors bind HES2-1 and 9 other transcription factors bind HES2-3 (out of 19 TFs analyzed). It is likely that the baseline H3K27ac at these sites is a result of the activity of these other transcription factors binding to these loci and recruiting cofactors. We now mention this in the results section:

Notably, at HES2-1 and HES2-3, there is significant baseline H3K27ac present, possibly due to the binding of other transcription factors to these sites (Table S2).

9) Fig4: why 2 algorithms? Why 500 parameter sets? Show the individual algorithms. Show the final convergence of the fit and the final correlation. The authors developed a thermodynamic model predicting the RNAPII binding (should correlate strongly with gene expression) to identify which interactions are crucial in the system. While this is an elegant model to explain the data, it should be described in a bit more details. For example, it is not clear why only best performed 500 parameter sets were chosen, why they have combined the results from 2 algorithms, what the bimodal distributions of most of the parameters mean.

We chose two algorithms, one based on gradient descent and one which does not require gradients, in order to verify that parameters were not biased by the algorithm chosen. We have now changed the selection of “good” parameter sets from the best 500 parameter sets to all sets which minimize the correlation with gene expression to within 0.1 of an optimal fit. These details have now been added to the results and methods section.

Results: We ran the parameter optimization with many random starts, then selected parameters which fit gene expression levels reasonably well (within 0.1 of optimal) (see Methods).

Methods: In order to control for potential bias resulting from the optimization algorithm used to fit the parameters, parameters were fit using both the L-BFGS-B algorithm (Byrd et al., 1995), which is a gradient descent-based method and the Hooke and Jeeves Pattern Search Optimization method (Hooke and Jeeves, 1961) which does not rely on gradient descent. An interface to these algorithms was implemented in R by John C Nash in the optimr package (Nash, 2016). Using both algorithms, 1000 iterations were run, resulting in 2000 parameter fits. We then selected parameter sets which correlated with gene expression data reasonably well (within 0.1 of an optimal correlation coefficient) for downstream analysis and plotting.

We have also added a mention of why we observe a bimodal distribution for some parameters in the results section:

In some cases, we observed multimodal parameter distributions, which is likely due to parameters balancing each other in different ways, leading to multiple local optima.

10) Global conclusion: is it the property of these enhancer regions that makes them work independently, or is it VP16? Can you do the model based on p300? Would that give insights into the mechanism at a different level (excluding the strong biases from the transactivator)

Please see our response to reviewer 2's general comment above.

Minor Points:

1) The introduction should be phrased more clearly. The necessity vs sufficiency concept could be explained more clearly (and earlier in the intro)

We appreciate the suggestion and have updated the introduction to read:

There are two different approaches to functionally perturb enhancers to study enhancer function. Deleting or inhibiting enhancer function tests the necessity of an enhancer for endogenous gene expression. The sufficiency of enhancer sequences can be studied by ectopic reporter assays (Catarino and Stark, 2018). However, testing only enhancer sequences does not uncover how enhancers act in their endogenous environment. To determine whether an enhancer region is sufficient within the genomic context, enhancers must be directly activated in an unbiased way.

2) S2D-F: show axis scales in the figure

This has been added.

Reviewer #3:

To the authors' credit, they acknowledge that the outcome of these two studies is fundamentally different in that enhancer-i uncovered clear synergistic enhancer behavior, whereas CRISPRa revealed mostly independent/additive behavior. Based on the provided data, this conclusion appears sound. However, what is then the major take home message of this study, beyond the rather technical observation that CRISPRa constructs, and especially the dCas9 one coupled to 10 (!) VP16 domains, are not useful to study enhancer function since they act as sledge hammer-like systems that override the subtler mechanisms of endogenous gene regulation? This is perhaps the most important issue of this study, since, as it stands, the offered conceptual advance is minor at best.

We acknowledge the reviewer's comments and as the reviewer points out, we discuss the limitations of the study in the discussion. However, we do not believe that the VP16(10x) is a sledge hammer (unless the reviewer thinks estrogen is a sledge hammer as well), since estrogen has similar effects on gene expression as discussed in our response to reviewer 2's general comment above. We have now also targeted individual sites with dCas9-p300(core) and created thermodynamic models of the data. The overall conclusion is the same when based on the less active dCas9-P300 synthetic activator (see reviewer 2's general comment). Overall, we think that while our study does have caveats, there are important advances, which we have outlined throughout the discussion and have summarized at the end of the discussion:

There are important caveats to consider when using these synthetic activators including the possible recruitment of cofactors that do not normally bind to a particular enhancer or potential interference with TF binding by dCas9. However, we believe that the comparison between enhancer activation and enhancer interference has shed light on the consistent importance of individual sites as well as a key difference in how enhancers work together when bound by different transcriptional activators.

Next to this major issue, there are several other aspects of this study that are only superficially covered even though their more in-depth, mechanistic analysis would significantly improve the paper's impact:

For example, to rationalize the differences between their two studies, the authors indicate that endogenous gene regulation is driven by a synergy between ERBSs, as mediated by the synergistic

recruitment of ER and its co-factors, in contrast to CRISPRa, which directly recruits activating cofactors to ERBSs. The authors write this as a factual statement, but no formal molecular evidence to support this statement and to more specifically demonstrate how these regulatory differences between the two utilized systems are molecularly encoded is provided. In other words, a big, question that this study does not resolve is why do ERBSs cooperate when targeted by ERa, but not when targeted by dCas9-VP16?

We agree with the reviewer that the big question that our study creates is why ER works synergistically while the synthetic activators do not. This is a very interesting, but also very open, question and we believe that it is motivation for future studies and outside the scope of the current manuscript. In the revised manuscript, we have shown that independence between enhancers is observed using dCas9-p300(core) (see our response to reviewer 2's general comment above), indicating that independence is not specific to dCas9-VP16(10x) and that synergy in the context of estrogen induction could be at the level of HAT recruitment and it is unlikely to be downstream.

The authors observe that dCas9-p300 activates genes to a lower level compared to dCas9-VP16, indicating that VP16 is likely contributing to gene activation "in other ways" (beyond mere H3K27Ac deposition) than p300. Here, the authors have an opportunity to go to the regulatory heart of co-factor function, yet they settle unfortunately at the observational rather than the mechanistic level.

We appreciate the suggestion and have looked at the differences between the two synthetic activators in more depth. We have now targeted dCas9-p300(core) to individual sites and found that the sites work together independently when targeted (see our response to reviewer 2's general comment above). The individual site targeting of dCas9-p300(core) also allowed us to explore the differences in activation when targeting each site. We found that activation by dCas9-VP16(10x) is associated with the binding of other transcription factors to the site, while dCas9-p300(core) activation is associated with higher baseline RNA polymerase II binding (data that was collected for this revision (see Figure S2H below). Please see our response to reviewer 1's 2nd comment for the added text and figures.

The presented data are seemingly at odds with the authors' statement that H3K27ac deposition by p300 and VP16 is similar. Indeed, as shown in Fig. 2C-D and Fig. S2D-E, the overall enrichment appears systematically lower for p300 compared to VP16 tracks. Fig. 2B summarizes the H3K27Ac data, showing that overall enrichment levels are comparable, but then how can the authors explain the provided track enrichment data, which clearly suggests otherwise?

We thank the reviewer for pointing this out. The original batch of H3K27ac ChIP-seq experiments had different levels of overall signal, which could be seen in the TBP promoter (non-targeted) in the original version of the manuscript. The ChIP tracks show non-normalized signal as we only normalized ChIP counts to total read depth. We have now done replicates of the ChIP-seq experiments as well as changed our normalization procedure and believe the results show similar H3K27ac. Please see our response to reviewer 2's 6th comment above, which includes a figure of the non-normalized data.

Does dCas9-p300 bind ERBSs to the same extent as dCas9-VP16? No dCas9-p300 ChIP-seq data is included in the manuscript so this question cannot be addressed based on the provided data. Also, why is the dCas9-VP16 ChIP data so variable and does this reflect technical or biological variation? If the latter, could this also influence the enhancer activation read-out?

We thank the reviewer for this suggestion and agree that it is an interesting question. We have now done replicate HA (dCas9) ChIP-seq experiments for both dCas9-VP16(10x) and dCas9-p300(core). We observed similar dCas9 recruitment for both fusions (see our response to reviewer 1's 1st comment above including the new Figure 2A). We have also done an analysis to determine if HA ChIP signal or H3K27ac signal are predictive of gene expression changes (see again our response to reviewer 1's 1st comment).

Finally, the authors show that sites that are adjacent to ERBSs but that are not targeted by ERa itself can also be targeted by CRISPRa, resulting in H3K27Ac deposition but not gene activation. Is this observation at odds with the statement that CRISPRa-targeted enhancers operate independently? In other words, an important and equally unresolved question is why these adjacent sites are unable to act as independent enhancers, even though they seem to be activated themselves based on H3K27Ac. Do the authors detect eRNAs for these sites and does their failure to participate in the gene activation process reflect local chromosome conformation? Answering these questions from a mechanistic rather than observational point of view would again be important to increase the conceptual value of the current study.

We agree that this is an interesting point. As the reviewer points out in a previous comment, the deposition of H3K27ac alone is not sufficient to activate gene expression based on the comparison of the two activating fusions and the analysis of adjacent regions. We found that the adjacent regions do not represent accessible chromatin and lack binding of other TFs and RNAPII. These observations go along with our analysis of predictors of activation by the synthetic activators (please see our response to reviewer 1's 2nd comment).

September 11, 2019

RE: Life Science Alliance Manuscript #LSA-2019-00497-TR

Dr. Jay Gertz
Huntsman Cancer Institute, University of Utah
Department of Oncological Sciences
2000 Circle of Hope
Salt Lake City, Utah 84112

Dear Dr. Gertz,

Thank you for submitting your revised manuscript entitled "Sufficiency analysis of estrogen responsive enhancers using synthetic activators". As you will see, the original reviewer #2 (here named 'reviewer #1') saw your manuscript again and thinks that the revision addresses the previously raised concerns well. We would thus be happy to publish your paper in Life Science Alliance pending final revisions necessary to meet our formatting guidelines:

- please link your ORCID iD to your profile in our submission system
- please fill in all mandatory fields in our submission system, including the summary blurb
- please move the legends of the supplementary figures into the main manuscript file and upload the main figures and suppl figures as individual files
- please provide your manuscript file in word docx format
- please upload the tables as excel or word files
- please include callouts in the manuscript text for Fig1A and Table S1

A. FINAL FILES:

B. MANUSCRIPT ORGANIZATION AND FORMATTING:

Sincerely,

Reviewer #1 (Comments to the Authors (Required)):

The authors have addressed all my comments to my satisfaction. The manuscript reads very well and the conclusions will be of broad interest in the gene regulation field.

September 18, 2019

RE: Life Science Alliance Manuscript #LSA-2019-00497-TRR

Dr. Jay Gertz
Huntsman Cancer Institute, University of Utah
Department of Oncological Sciences
2000 Circle of Hope
Salt Lake City, Utah 84112

Dear Dr. Gertz,

Thank you for submitting your Research Article entitled "Sufficiency analysis of estrogen responsive enhancers using synthetic activators". It is a pleasure to let you know that your manuscript is now accepted for publication in Life Science Alliance. Congratulations on this interesting work.

*****IMPORTANT:** If you will be unreachable at any time, please provide us with the email address of an alternate author. Failure to respond to routine queries may lead to unavoidable delays in publication.*******

DISTRIBUTION OF MATERIALS:

Again, congratulations on a very nice paper. I hope you found the review process to be constructive and are pleased with how the manuscript was handled editorially. We look forward to future exciting submissions from your lab.

Sincerely,
